# MrsFormer: Transformer with Multiresolution-head Attention

## Abstract

We propose the Transformer with Multiresolution-head Attention (MrsFormer), a class of efficient transformers inspired by the multiresolution approximation (MRA) for approximating a signal $f$ using wavelet bases. MRA decomposes a signal into components that lie on orthogonal subspaces at different scales. Similarly, MrsFormer decomposes the attention heads in the multi-head attention into fine-scale and coarse-scale heads, modeling the attention patterns between tokens and between groups of tokens. Computing the attention heads in MrsFormer requires significantly less computation and memory footprint compared to the standard softmax transformer with multi-head attention. We analyze and validate the advantage of MrsFormer over the standard transformers on a wide range of applications including image and time series classification.

## 1 Introduction

The transformer architectures (Vaswani et al., 2017) is popularly used in natural language processing (Devlin et al., 2018; Al-Rfou et al., 2019; Dai et al., 2019; Child et al., 2019; Raffel et al., 2020; Baevski & Auli, 2019; Brown et al., 2020; Dehghani et al., 2018), computer vision (Dosovitskiy et al., 2021; Liu et al., 2021; Touvron et al., 2020; Ramesh et al., 2021; Radford et al., 2021; Arnab et al., 2021; Liu et al., 2022; Zhao et al., 2021; Guo et al., 2021; Chen et al., 2022), speech processing (Gulati et al., 2020; Dong et al., 2018; Zhang et al., 2020; Wang et al., 2020b), and other relevant applications (Rives et al., 2021; Jumper et al., 2021; Chen et al., 2021; Zhang et al., 2019; Wang & Sun, 2022). Transformers achieve state-of-the-art performance in many of these practical tasks, and the results get better with larger model size and increasingly long sequences. For example, the text generating model in (Liu et al., 2018a) processes input sequences of up to 11,000 tokens of text. Applications involving other data modalities, such as music (Huang et al., 2018) and images (Parmar et al., 2018), can require even longer sequences. Lying at the heart of transformers is the self-attention mechanism, an inductive bias that connects each token in the input through a relevance weighted basis of every other tokens to capture the contextual representation of the input sequence (Cho et al., 2014; Parikh et al., 2016; Lin et al., 2017; Bahdanau et al., 2014; Vaswani et al., 2017; Kim et al., 2017). The capability of self-attention to attain diverse syntactic and semantic representations from long input sequences accounts for the success of transformers in practice (Tenney et al., 2019; Vig & Belinkov, 2019; Clark et al., 2019; Voita et al., 2019a; Hewitt & Liang, 2019). The multi-head attention (MHA) extends the self-attention by concatenating multiple attention heads to compute the final output as explained in Section 2.1 below.

In spite of the success of the MHA, it has been shown that attention heads in MHA are redundant and tend to learn similar attention patterns, thus limiting the representation capacity of the model (Michel et al., 2019; Voita et al., 2019b; Bhojanapalli et al., 2021). Furthermore, additional heads increase the computational and memory costs, which becomes a bottleneck in scaling up transformers for very long sequences in large-scale practical tasks. These high computational and memory costs and head redundancy issues of the MHA motivates the need for a new efficient attention mechanism.

### 1.1 Contribution

Leveraging the idea of the multiresolution approximation (MRA) (Mallat, 1999; 1989; Crowley, 1981), we propose a class of efficient and flexible transformers, namely the Transformer with Multiresolution-head Attention (MrsFormer). At the core of MrsFormer is to use the novel Multiresolution-head Attention (MrsHA) that computes the approximation of the outputs $\mathbf{H}^h$, $h = 1, \ldots, H$, of attention heads in MHA at different scales for saving computation and reducing the memory cost of the

model. The MRA has been widely used to efficiently approximate complicated signals like video and images in signal and image processing (Mallat, 1999; Taubman & Marcellin, 2002; Bhaskaran & Konstantinides, 1997), as well as to approximate solutions of partial differential equations (Dahmen et al., 1997; Qian & Weiss, 1993). While existing works have been proposed to approximate the attention matrices using the MRA (Zeng et al., 2022; Fan et al., 2021; Tao et al., 2020; Li et al., 2022), our MrsHA is the first method that approximates the output of an attention head, resulting in a better approximation scheme compared to other works that try to approximate the attention matrices. Our contribution is three-fold:

1. We derive the approximation of an attention head at different scales via two steps: i) Directly approximating the output sequence $\mathbf{H}$, and ii) approximating the value matrix $\mathbf{V}$, i.e. the dictionary that contains bases of $\mathbf{H}$.

2. We develop MrsHA, a novel MHA whose attention heads approximate the output sequences $\mathbf{H}^h$, $h = 1, \ldots, H$, at different scales. We then propose MrsFormer, a new class of transformers that use MrsHA in their attention layers.

3. We empirically verify that the MrsFormer helps reduce the head redundancy and achieves better efficiency than the baseline softmax transformer while attaining comparable accuracy to the baseline.

**Organization:** We structure this paper as follows: In Section 2, we derive the approximation for the output sequence $\mathbf{H}^h$, $h = 1, \ldots, H$, at different scales and propose the MrsHA and MrsFormer. In Section 3 and 4, we empirically validate and analyze the advantages of the MrsFormer over the baseline softmax transformer. We discuss related work in Section 5. The paper ends up with concluding remarks. More experimental details are provided in the Appendix.

## 2 TRANSFORMER WITH MULTIRESOLUTION-HEAD ATTENTION

### 2.1 BACKGROUND: SELF-ATTENTION

The self-attention mechanism learns long-range dependencies via parallel processing of the input sequence. For a given input sequence $\mathbf{X} := [\boldsymbol{x}_1, \cdots, \boldsymbol{x}_N]^\top \in \mathbb{R}^{N \times D_x}$ of $N$ feature vectors, the self-attention transforms $\mathbf{X}$ into the output sequence $\mathbf{H} := [\boldsymbol{h}_1, \cdots, \boldsymbol{h}_N]^\top \in \mathbb{R}^{N \times D_v}$ as follows

$$\mathbf{H} = \mathrm{softmax}\Big(\frac{\mathbf{QK}^\top}{\sqrt{D}}\Big)\mathbf{V} := \mathbf{AV}, \tag{1}$$

where $\boldsymbol{Q} := [\boldsymbol{q}_1, \cdots, \boldsymbol{q}_N]^\top, \mathbf{K} := [\boldsymbol{k}_1, \cdots, \boldsymbol{k}_N]^\top$, and $\mathbf{V} := [\boldsymbol{v}_1, \cdots, \boldsymbol{v}_N]^\top$ are the projections of the input sequence $\mathbf{X}$ into three different subspaces spaned by $\mathbf{W}_Q, \mathbf{W}_K \in \mathbb{R}^{D \times D_x}$, and $\mathbf{W}_V \in \mathbb{R}^{D_v \times D_x}$, i.e. $\mathbf{Q} = \mathbf{XW}_Q^\top, \mathbf{K} = \mathbf{XW}_K^\top, \mathbf{V} = \mathbf{XW}_V^\top$. Here, in the context of transformers, $\mathbf{Q}, \mathbf{K}$, and $\mathbf{V}$ are named the query, key, and value matrices, respectively. The softmax function is applied to row-wise. The matrix $\mathbf{A} = \mathrm{softmax}\Big(\frac{\mathbf{QK}^\top}{\sqrt{D}}\Big) \in \mathbb{R}^{N \times N}$ is the attention matrix, whose component $a_{ij}$ for $i, j = 1, \cdots, N$ are the attention scores. The structure of the attention matrix $\mathbf{A}$ after training from data determines the ability of the self-attention to capture contextual representation for each token. Eqn. (1) is also called the scaled dot-product or softmax attention. In our paper, we call a transformer that uses this attention the softmax transformer.

**Multi-head Attention (MHA).** In MHA, multiple heads are concatenated to compute the final output. Let $H$ be the number of heads and $\mathbf{W}_O^{\mathrm{multi}} = \Big[\mathbf{W}_O^{(1)}, \ldots, \mathbf{W}_O^{(H)}\Big] \in \mathbb{R}^{D_v \times HD_v}$ be the projection matrix for the output where $\mathbf{W}_O^{(1)}, \ldots, \mathbf{W}_O^{(H)} \in \mathbb{R}^{D_v \times D_v}$. The multi-head attention is defined as

$$\mathrm{MultiHead}(\{\mathbf{H}\}_{h=1}^H) = \mathrm{Concat}(\mathbf{H}^{(1)}, \ldots, \mathbf{H}^{(H)})\mathbf{W}_O^{\mathrm{multi}\top}$$

$$= \sum_{h=1}^H \mathbf{H}^{(h)}\mathbf{W}_O^{h\top} = \sum_{h=1}^H \mathbf{A}^{(h)}\mathbf{V}^{(h)}\mathbf{W}_O^{(h)\top}. \tag{2}$$

The MHA enables transformers to capture more diverse attention patterns.

### 2.2 BACKGROUND: WAVELET TRANSFORM AND MULTIRESOLUTION APPROXIMATIONS

The wavelet transform uses time-frequency atoms with different time supports to analyze the structure of a signals. In particular, it decomposes signals over dilated and translated copies of a fixed function

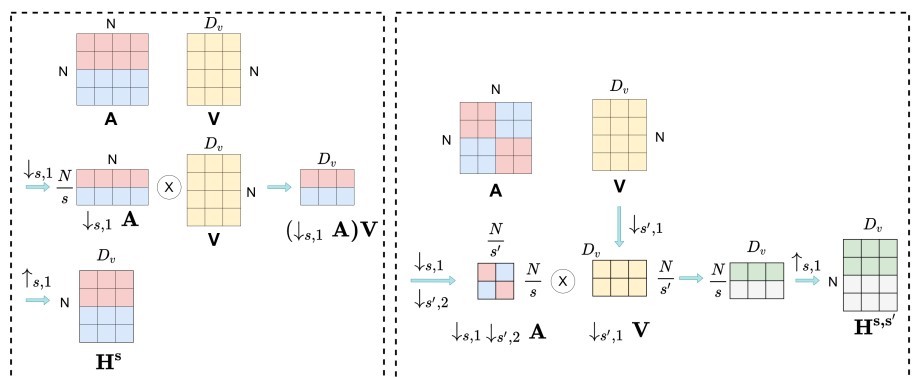

Figure 1: Illustration of Eqn. 11 (Left) and Eqn. 14 (Right).

$\varphi$. A dictionary of time-frequency atoms is obtained by scaling $\varphi$ by $s$ and translating it by $t$:

$$\mathcal{B} = \left\{ \varphi_t^s = \frac{1}{\sqrt{s}} \varphi \left( \frac{x-t}{s} \right) \right\}_{t \in \mathbb{R}, s \in \mathbb{R}^+}. \tag{3}$$

Here, $s$ controls the dilation, i.e., the scale, and $t$ controls the location, e.g., the time. Using this dictionary of time-frequency atoms, a signal $f \in \mathbf{L}^2(\mathbb{R})$ can be expanded in the following form:

$$f = \int_0^{+\infty} \int_{-\infty}^{+\infty} \alpha_t^s \varphi_t^s(x) \, dt ds. \tag{4}$$

The wavelet transform then maps the signal $f$ to the coefficient $\alpha_t^s$ as follows

$$\alpha_t^s = \langle f, \varphi_t^s \rangle = \int_{-\infty}^{+\infty} f(x)(\varphi^*)_t^s \, dx, \tag{5}$$

where $\varphi^*$ is the complex conjugate of $\varphi$. The coefficient $\alpha_t^s$ captures the measurement of the signal $f$ at scale $s$ and location $t$ (Mallat, 1999).

## 2.3 TRANSFORMER WITH MULTIRESOLUTION-HEAD ATTENTION

### 2.3.1 FIRST LEVEL APPROXIMATION: APPROXIMATING THE OUTPUT SEQUENCE $\mathbf{H}$ AT DIFFERENT SCALES

Let $\mathcal{B}^s = \{\varphi_t^s \in \mathbb{R}^N\}$ be a set of orthogonal expansion functions for possible translations at scale $s$ where $s = 1, 2, 4, \ldots, N$. For simplicity, we assume that the sequence length $N = 2^k$. The expansion functions $\varphi_t^s$ are chosen to be the boxcar functions as follows

$$\varphi_t^s[i] = \begin{cases} 1 & \text{if } st - s < i \leq st \\ 0 & \text{otherwise} \end{cases} \tag{6}$$

for $s \in \{1, 2, 4, \ldots, N\}$ and $t \in \{1, \ldots, N/s\}$. At each scale $s$, we approximate the columns $\mathbf{H}[:, d]$, $d = 1, \ldots, D_v$, of the output sequence $\mathbf{H}$ as follows

$$\mathbf{H}[:, d] \approx \mathbf{H}^s[:, d] = \sum_{\varphi_t^s \in \mathcal{B}^s} \alpha_{td}^s \varphi_t^s, \tag{7}$$

where the coefficient $\alpha_{td}^s$ is computed as follows

$$\alpha_{td}^s = \frac{1}{s} \langle \varphi_t^s, \mathbf{H}[:, d] \rangle. \tag{8}$$

Plug Eqn. (1) and Eqn. (8) into Eqn. (7), we obtain

$$\mathbf{H}[:, d] \approx \mathbf{H}^s[:, d] = \sum_{\varphi_t^s \in \mathcal{B}^s} \frac{1}{s} \langle \varphi_t^s, \mathbf{H}[:, d] \rangle \varphi_t^s = \sum_{t=1}^{N/s} \left( \frac{1}{s} \sum_{i=st-s+1}^{st} \mathbf{H}[i, d] \right) \varphi_t^s$$

$$= \sum_{t=1}^{N/s} \left( \left( \frac{1}{s} \sum_{i=st-s+1}^{st} \mathbf{A}[i, :] \right) \mathbf{V}[:, d] \right) \varphi_t^s \tag{9}$$

$$= \uparrow_{s,1} \left( (\downarrow_{s,1} \mathbf{A}) \mathbf{V}[:, d] \right). \tag{10}$$

Here, we employ the notations for downsampling and upsampling from signal processing. In particular, $\downarrow_{s,\ell}$ denotes the average pooling by the factor $s$ along the $\ell^{th}$ dimension, and $\uparrow_{s,\ell}$ denotes the nearest-neighbor interpolation by the factor $s$ along the $\ell^{th}$ dimension. Applying Eqn. (10) for $d = 1, \ldots, D_v$, we achieve the approximation of $\mathbf{H}$ at scale $s$ as follows:

$$\mathbf{H} \approx \mathbf{H}^s = \uparrow_{s,1} ((\downarrow_{s,1} \mathbf{A})\mathbf{V}). \tag{11}$$

An illustration of Eqn. 11 is given in Figure. 1 (Left).

**Remark 1 (Approximating the columns of $\mathbf{H}$ independently)** *As pointed out in (Nguyen et al., 2022), the features $\mathbf{H}[:,d]$ in the ouput sequence $\mathbf{H}$, as well as the features $\mathbf{V}[:,d]$ in the value matrix $\mathbf{V}$, $d = 1, \ldots, D_v$, in the softmax attention are independent due to the use of the unnormalized Gaussian kernels with the isotropic covariance. This finding justifies our approach of approximating the columns of $\mathbf{H}$ independently.*

**Remark 2 (Group-to-token attention)** *The downsampling $\downarrow_{s,1} \mathbf{A}$ of the matrix $\mathbf{A}$ in Eqn. (11) computes the attentions between groups of tokens and individual tokens in the sequence.*

### 2.3.2 SECOND LEVEL APPROXIMATION: APPROXIMATING THE HEAD BASES $\mathbf{V}$ AT DIFFERENT SCALES

In Eqn. (11) that approximates the output sequence $\mathbf{H}$ at scale $s$ by $\mathbf{H}^s$, we can further approximate the bases $\mathbf{V}$, i.e., the value matrix, by its approximation at scale $s'$. Following the derivation in Section 2.3.1 above, we can derive the approximation $\mathbf{V}^{s'}[:,d]$ for the $d^{th}$ columns of $\mathbf{V}$ as follows

$$\mathbf{V}[:,d] \approx \mathbf{V}^{s'}[:,d] = \sum_{t'=1}^{N/s'} \left( \frac{1}{s'} \sum_{j=s't'-s'+1}^{s't'} \mathbf{V}[j,d]) \right) \boldsymbol{\varphi}_{t'}^{s'}. \tag{12}$$

Plugging Eqn. (12) into Eqn. (9), we obtain the second level approximation of the head output $\mathbf{H}$:

$$\mathbf{H}[:,d] \approx \mathbf{H}^{s,s'}[:,d]$$

$$= \sum_{t=1}^{N/s} \left( \left( \frac{1}{s} \sum_{i=st-s+1}^{st} \mathbf{A}[i,:] \right) \sum_{t'=1}^{N/s'} \left( \frac{1}{s'} \sum_{j=s't'-s'+1}^{s't'} \mathbf{V}[j,d]) \right) \boldsymbol{\varphi}_{t'}^{s'} \right) \boldsymbol{\varphi}_t^s$$

$$= \sum_{t=1}^{N/s} \left( \sum_{t'=1}^{N/s'} \left( \frac{1}{s's} \sum_{i=st-s+1}^{st} \mathbf{A}[i,:]\boldsymbol{\varphi}_{t'}^{s'} \right) \left( \sum_{j=s't'-s'+1}^{s't'} \mathbf{V}[j,d]) \right) \right) \boldsymbol{\varphi}_t^s$$

$$= \sum_{t=1}^{N/s} \left( \sum_{t'=1}^{N/s'} \left( \frac{1}{s's} \sum_{i=st-s+1}^{st} \sum_{j=s't'-s'+1}^{s't'} \mathbf{A}[i,j] \right) \left( \sum_{j=s't'-s'+1}^{s't'} \mathbf{V}[j,d]) \right) \right) \boldsymbol{\varphi}_t^s$$

$$= \uparrow_{s,1} ((\downarrow_{s,1}\downarrow_{s',2} \mathbf{A})(\downarrow_{s',1} \mathbf{V}[:,d])). \tag{13}$$

Same as above, by applying Eqn. (13) for $d = 1, \ldots, D_v$, we achieve the full approximation of $\mathbf{H}$ at scale $s$ of $\mathbf{H}$ and scale $s'$ of $\mathbf{V}$ as follows:

$$\mathbf{H} \approx \mathbf{H}^{s,s'} := \uparrow_{s,1} ((\downarrow_{s,1}\downarrow_{s',2} \mathbf{A})(\downarrow_{s',1} \mathbf{V})). \tag{14}$$

An illustration of Eqn. 14 is given in Figure. 1 (Right). Given the approximation $\mathbf{H}^{s,s'}$ of the attention matrix $\mathbf{H}$, we have the following upper bound on the approximation error.

**Theorem 1** *Assume that $\delta > 0$ is chosen such that the attention matrix $\mathbf{A}$ satisfies the following inequalities $|\mathbf{A}_{i,j} - \mathbf{A}_{i\pm1,j}| \leq \delta$, $|\mathbf{A}_{i,j} - \mathbf{A}_{i,j\pm1}| \leq \delta$ for all $1 \leq i, j \leq N$. Then, we obtain that*

$$\|\mathbf{H} - \mathbf{H}^{s,s'}\|_F \leq \frac{(s + s' - 2)N\delta}{\sqrt{ss'}}\|\mathbf{V}\|_2,$$

*where $\|.\|_F$ denotes the Frobenius norm and $\|.\|_2$ denotes the spectral norm of a matrix.*

Proof of Theorem 1 is in Appendix B. The result of Theorem 1 shows that the approximation matrix $\mathbf{H}^{s,s'}$ approximates $\mathbf{H}$ exactly when $s = s' = 1$, which is true. In the coarsest scale when $s = s' = N$, the upper bound achieves the maximum value $(N - 1)\delta\|\mathbf{V}\|_2$.

**Remark 3 (Group-to-group attention)** *The downsampling $\downarrow_{s,1}\downarrow_{s',2} \mathbf{A}$ of the matrix $\mathbf{A}$ in Eqn. (14) computes the attentions between groups of tokens and groups of tokens in the sequence.*

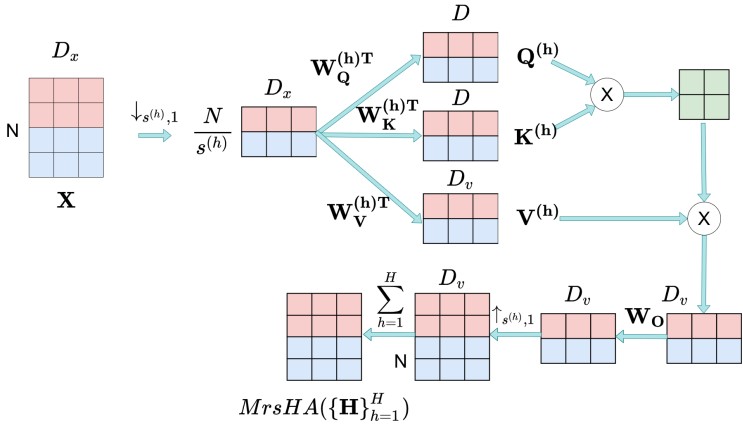

Figure 2: Illustration of Eqn. 17.

### 2.3.3 Efficient Downsampling of the Attention Matrix $\mathbf{A}$

As shown in Eqn. (1), $\mathbf{A} = \text{softmax}\left(\frac{\mathbf{Q}\mathbf{K}^\top}{\sqrt{D}}\right)$. Since the softmax function needs access to the full matrix $\frac{\mathbf{Q}\mathbf{K}^\top}{\sqrt{D}}$, downsampling $\mathbf{A}$ via average pooling still requires to compute the full product $\frac{\mathbf{Q}\mathbf{K}^\top}{\sqrt{D}}$ first. In order to avoid this redundant computation, we propose to compute the lower bound of this average pooling (due to the convexity of the exponential in the softmax function). In particular, we approximate the downsampling of $\mathbf{A}$ as follows:

$$\downarrow_{s,1}\downarrow_{s',2} \mathbf{A} \approx \text{softmax}\left(\frac{\downarrow_{s,1}\downarrow_{s',2}(\mathbf{Q}\mathbf{K}^\top)}{\sqrt{D}}\right) = \text{softmax}\left(\frac{(\downarrow_{s,1}\mathbf{Q})(\downarrow_{s',1}\mathbf{K})^\top}{\sqrt{D}}\right). \quad (15)$$

### 2.3.4 Transformer with Multiresolution-head Attention: Each Head Approximates the Attention at a Different Scale

In this section, we formally define our Multiresolution-head Attention (MrsHA) and Transformer with Multiresolution-head Attention (MrsFormer). MrsHA combines Eqn. (14) and (15) to implement the approximation of the output sequences $\mathbf{H}^{(h)}$, $h = 1, \ldots, H$, at different scales $s$ and $s'$.

**Definition 1 (Multiresolution-head Attention)** *Let $H$ be the number of heads and $\mathbf{W}_O^{multi} = \left[\mathbf{W}_O^{(1)}, \ldots, \mathbf{W}_O^{(H)}\right] \in \mathbb{R}^{D_v \times HD_v}$ be the projection matrix for the head outputs where $\mathbf{W}_O^{(1)}, \ldots, \mathbf{W}_O^{(H)} \in \mathbb{R}^{D_v \times D_v}$. Given a set of scales $\{s^{(h)}, s'^{(h)}\}_{h=1}^H$ for the output $\mathbf{H}^{(h)}$ and the value matrix $\mathbf{V}^{(h)}$, $h = 1, \ldots, H$, at each head, the MrsHA is an efficient attention mechanism that computes the approximation of $\mathbf{H}^{(h)}$ at scale $s^{(h)}$ using an approximation of $\mathbf{V}^{(h)}$ at scale $s'^{(h)}$ by the following attention formula:*

$$MrsHA(\{\mathbf{H}\}_{h=1}^H) = \sum_{h=1}^H \uparrow_{s^{(h)},1}\left(\text{softmax}\left(\frac{(\downarrow_{s^{(h)},1}\mathbf{Q})(\downarrow_{s'^{(h)},1}\mathbf{K})^\top}{\sqrt{D}}\right)(\downarrow_{s'^{(h)},1}\mathbf{V}^{(h)})\right)\mathbf{W}_O^{(h)\top}. \quad (16)$$

*The MrsFormer is the class of transformers that use the MrsHA in their attention layers.*

**Remark 4 (Downsampling $\mathbf{Q}$, $\mathbf{K}$, and $\mathbf{V}$)** *Downsampling $\mathbf{Q}$, $\mathbf{K}$, and $\mathbf{V}$ can be efficiently implemented by downsampling the input sequence $\mathbf{X}$ before projecting it into the query matrix $\mathbf{Q}$, the key matrix $\mathbf{K}$, and the value matrix $\mathbf{V}$ via the linear transformations $\mathbf{W}_Q$, $\mathbf{W}_K$, and $\mathbf{W}_V$, respectively. Eqn. (16) of the MrsHA then becomes*

$$MrsHA(\{\mathbf{H}\}_{h=1}^H)$$
$$= \sum_{h=1}^H \uparrow_{s^{(h)},1}\left(\text{softmax}\left(\frac{(\downarrow_{s^{(h)},1}\mathbf{X}\mathbf{W}_Q^{(h)\top})(\downarrow_{s'^{(h)},1}\mathbf{X}\mathbf{W}_K^{(h)\top})^\top}{\sqrt{D}}\right)(\downarrow_{s'^{(h)},1}\mathbf{X}\mathbf{W}_V^{(h)\top})\right)\mathbf{W}_O^{(h)\top}. \quad (17)$$

*An illustration of Eqn. 17 is given in Figure. 2.*

**Remark 5 (Choosing $s^{(h)}$ and $s'^{(h)}$)** $s^{(h)}$ and $s'^{(h)}$ are hyperparameters that can be tuned for each head. In our experiments, we use $s^{(h)} = s'^{(h)} = 2^{k^{(h)}}$, where $k^{(h)}$ is an integer.

**Remark 6 (Choosing the expansion functions $\varphi_t^s$ and 1-D convolution)** *In order to derive the MrsHA in Eqn. (16), we have chosen the expansion functions $\varphi_t^s$ to be the boxcar functions. Other expansion functions, such as the wavelet bases or the triangular functions, can be used to derive different forms of the MrsHA. In a general case, the average pooling and the nearest-neighbor interpolation in Eqn. (16) and (17) can be replaced by the 1-D convolution operators with $\varphi_t^s$ as the corresponding filters.*

## 3 EXPERIMENTAL RESULTS

In this section, we empirically justify the advantages of our propsed MrsFormer model. We compare the performance of the MrsFormer with the baseline softmax transformer, the MRA-2 (Zeng et al., 2022), and the MRA-2-s (which is the sparse version of the MRA-2) on various benchmarks. Unlike our method, the MRA-2 and MRA-2-s perform multiresolution analysis for each head by approximating the attention matrix by blocks of different scales, while the MrsHA in our MrsFormer computes the approximation of each head $\mathbf{H}^h$ at a specific scale. The benchmarks studied in our experiments include 10 tasks from the UEA time series classification dataset (Bagnall et al., 2018), 3 tasks from Long Range Arena (Tay et al., 2021b) (LRA) benchmark, and ImageNet image classification task (Russakovsky et al., 2015). In addition, we also study the performance of the MrsHA when being combined with other attention mechanism such as the linear attention (Katharopoulos et al., 2020), the MRA-2 attention, and the MRA-2-s attention (Zeng et al., 2022). We aim to show that: (i) the MrsFormer can achieve better or comparable accuracy over the baseline softmax, MRA-2, and MRA-2-s transformers; (ii) the MrsFormer saves significant amount of FLOPs and memory compared to the baseline softmax transformer, and this advantage grows with the sequence length; (iii) the MrsHA can be combined with other attentions to achieve similar or better performance with better efficiency; and (iv) the MrsFormer reduces redundancy between heads comparing to the softmax baseline.

In our experiment, we keep the hyperparameters the same for all models for fair comparisons. All of our results are averaged over 5 runs with different seeds.

### 3.1 UEA TIME SERIES CLASSIFICATION

**Models and baselines.** We adapt code from (Wu et al., 2022; Zerveas et al., 2021) for our experiments. Following the same setting from these papers, we set the number of heads and layers to 8 and 2, respectively. For the MrsFormers, we use the same set of scales at each layer, which is given by $\boldsymbol{s} = [1, 1, 2, 2, 4, 4, 8, 8]$. For MRA-2 and MRA-2-s models (Zeng et al., 2022), each head is approximated by blocks of scales $[1, 32]$ as suggested in their paper. The percentage of blocks with scale 1 in these MRA-2 models is set to $25\%$ of the full attention matrix. Other hyperparameters have the same values as in (Wu et al., 2022) (for the PEMS-SF, SelfRegulationSCP2, and UWaveGestureLibrary tasks) and (Zerveas et al., 2021) (for other tasks).

**Results.** We summarize the results in Table 1. The MrsFormer achieves bettter test accuracy than the baseline softmax transformer for 5 out of 10 tasks while being much more efficient. Among these tasks, the MrsFormer outperforms the baseline by at least 1% accuracy. For the remaining tasks, besides Handwriting, our model maintains an accuracy gap less than 0.8% compared to the baseline. Our model gets the best accuracy for 4 out of the 10 tasks. In addition, it achieves second best accuracy for 4 out of the remaining tasks. The MrsFormer achieves the average accuracy across all tasks. Note that among 8 heads at each layer, our model computes 6 of them with the size of only $\frac{1}{4}, \frac{1}{4}, \frac{1}{16}, \frac{1}{16}, \frac{1}{64}$ and $\frac{1}{64}$ of the size of the corresponding heads in the baseline softmax transformer. Thus, the MrsFormer has a significant smaller FLOPS and memory usage compared to the baseline.

### 3.2 LONG RANGE ARENA

**Models and baselines.** We follow the same settings and adapt code for LRA task from (Zeng et al., 2022), which uses transformer with 2 heads and 2 layers. We choose the same set of scales $\boldsymbol{s} = [1, 2]$ for all the layers in MsFormer.

**Results.** Table 2 summarizes our results. Although being an approximation of the softmax attention, it is evidently from Table 2 that MrsFormer can consistently achieve better than or comparable accuracy as the baseline softmax attention on the LRA tasks. The MRA-2 and MRA-2-s models (Zeng et al.,

Table 1: Accuracy (%) of the MrsFormer vs. the baseline softmax transformer on the UEA Time Series Classification task averaged over 5 seeds. The best model for each task is highlighted in bold, while the second best one is underlined. We also include the reported results for the softmax transformer from (Wu et al., 2022) and (Zerveas et al., 2021) (in parentheses). The MrsFormer attains the best average accuracy across all tasks while being much more efficient than the baseline softmax transformers.

| DATASET / MODEL | BASELINE SOFTMAX | MRSFORMER | MRA-2 | MRA-2-S |
|---|---|---|---|---|
| ETHANOLCONCENTRATION | 32.08 (33.70) | **35.87** | 34.35 | 34.48 |
| FACEDETECTION | **68.70** (68.10) | 68.23 | 68.28 | 68.24 |
| HANDWRITING | **32.08** (30.50) | 30.24 | 29.49 | 29.68 |
| HEARTBEAT | 75.77 (77.60) | **78.86** | 77.24 | 78.05 |
| JAPANESEVOWELS | **99.46** (99.40) | 99.10 | 99.01 | 99.01 |
| PEMS-SF | 82.66 (82.10) | 84.2 | **86.13** | 82.85 |
| SELFREGULATIONSCP1 | 91.46 (**92.50**) | 91.81 | 91.70 | 92.04 |
| SELFREGULATIONSCP2 | 54.72 (53.90) | **56.85** | 55.56 | 56.29 |
| SPOKENARABICDIGITS | **99.33** (99.30) | 98.73 | 98.60 | 98.62 |
| UWAVEGESTURELIBRARY | 84.45 (85.60) | **86.67** | **86.67** | 86.56 |
| AVERAGE ACCURACY | 72.07(72.27) | **73.06** | 72.70 | 72.58 |

Table 2: Accuracy (%) of the MrsFormer vs. the baseline softmax transformer averaged over 5 seeds. The best model for each task is highlighted in bold, while the second best one is underlined. The MrsFormer attains the best average accuracy across all tasks while being much more efficient than the baseline softmax transformers.

| DATASET / MODEL | BASELINE SOFTMAX | MRSFORMER | MRA-2 | MRA-2-S |
|---|---|---|---|---|
| LISTOPS | 36.84 (37.10) | **37.52** | 37.10 (37.2) | 37.05 (37.4) |
| RETRIEVAL | 79.52 (79.6) | **80.22** | 78.88 (79.6) | 79.76 (80.3) |
| TEXT | 64.93 (65.2) | 65.05 | **65.09** (65.4) | 64.43 (64.3) |
| AVERAGE ACCURACY | 60.43 (60.63) | **60.93** | 60.36 (60.73) | 60.41 (60.67) |

Table 3: Accuracy (%) of the MrsFormer DeiT vs. the baseline softmax DeiT and the MRA-2-s DeiT on the ImageNet image classification task. The MrsFormer DeiT outperforms the MRA-2-s DeiT and yields comparable accuracy to the softmax DeiT.

| MODEL NAME | TOP-1 ACCURACY | TOP-5 ACCURACY |
|---|---|---|
| SOFTMAX DEIT | 72.178 | 91.126 |
| MRA-2-S DEIT | 70.784 | 90.154 |
| MRSFORMER DEIT | 71.342 | 90.566 |

2022) are also included for comparison. Our MrsFormer's performance is comparable with these MRA baselines. Overall, the MrsFormer yields the best average accuracy across the LRA tasks.

### 3.3 IMAGENET

**Models and baselines:** In this section, we apply the MrsFormer to the Deit model (Touvron et al., 2020) with 4 heads. Since Deit uses special class token $[CLS]$ for the classification, we do not downsample this token along with other tokens in the sequence. For our MrsFormers, we use the set of scales $s = [1, 2, 2, 4]$ at each layer. We also study the MRA-2-s attention on this task. As reported in (Zeng et al., 2022), the MRA-2-s is a better model than the MRA-2 on the ImageNet image classification task since its sparse attention structure is more effective for modeling images.

**Results:** We present our results in Table 3. The MrsFormer DeiT's top-1 accuracy is about $0.5\%$ higher than MRA-2-s DeiT and is the closest model to the performance of the softmax DeiT baseline. The performance gap of less than $1\%$ of MrsFormer DeiT is very promising for applying the MrsFormer-based model in large scale tasks to reduce the computational and memory cost while maintaining comparable performance with the baseline transformer.

## 4 EMPIRICAL ANALYSIS

In this section, we use the models trained on the LRA retrieval task for our analysis.

### 4.1 EFFICIENCY ANALYSIS

We study the efficiency of MrsFormer over the baseline softmax transformer. Figure 3 demonstrates the reduction ratio of train and test flops of the MrsFormer over the softmax transformer. Although

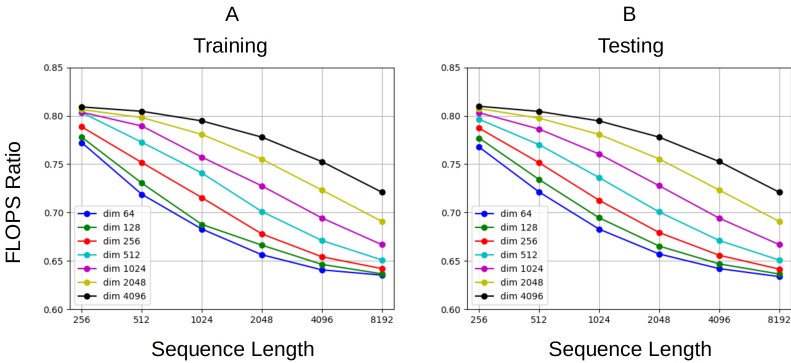

Figure 3: Training (A) and inference (B) FLOP ratios between the MrsFormer and the baseline softmax transformer across different model dimensions $D$ (dim) and sequence lengths $N$ on the LRA retrieval task. The MrsFormer requires fewer FLOPs compared to the baseline, and this advantage grows with the sequence length for very long sequences. The efficient advantage of the MrsFormer holds for large-scale models with the large $D$.

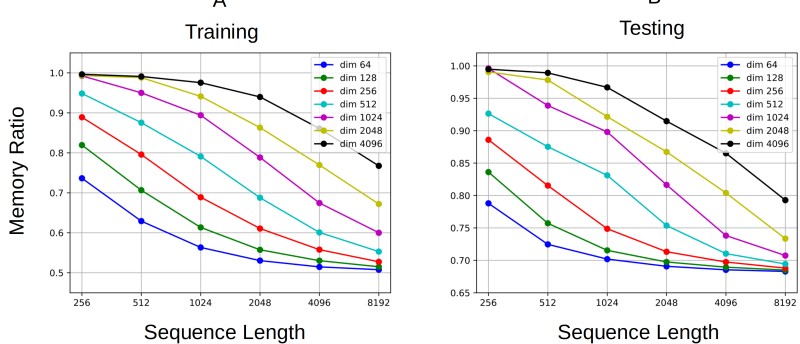

Figure 4: Training (A) and inference (B) memory ratios between the MrsFormer and the baseline softmax transformer across different model dimensions $D$ (dim) and sequence lengths $N$ on the LRA retrieval task.

Table 4: Layer-average mean and standard deviation of $\mathcal{L}_2$ distances between heads of the MrsFormer vs. the softmax transformer trained on the retrieval task. The MrsFormer obtains greater $\mathcal{L}_2$ distances between heads compared to the baseline, indicating that the MrsFormer captures more diverse attention patterns.

| MetricModel | Baseline Softmax | MrsFormer |
|---|---|---|
| Mean | 2.01 | **2.68** |
| Std | 0.39 | **0.54** |

in this experiment, we only approximate one head with scale $s = 2$ and preserve the other head the same as in the baseline, the FLOP saving ratio over softmax attention still ranges from 18% up to more than 36% and grows with sequence length in both the training and testing phases. Figure 4 presents the memory saving ratio of the MrsFormer over the softmax transformer. This figure shows a similar trend of more memory saving when the sequence length increases. Our model achieves up to 49% and 31% decrease in memory usage in the training and testing phases, respectively. This indicates that our model scales well with long sequences and takes significantly less resource than the baseline softmax attention in both training and testing.

### 4.2 MRSFORMER HELPS REDUCE HEAD REDUNDANCY

To show that the MrsFormer captures more diverse attention patterns, we compare the average $\mathcal{L}_2$ distances between the heads of our trained MrsFormer model (on the retrieval task) and the softmax baseline. Table 4 reports the layer-average mean and standard deviation of distances between heads. Since the MrsFormer attains higher $\mathcal{L}_2$ distances, it reduces the risk of learning redundant heads compared to the softmax baseline.

### 4.3 BEYOND THE SOFTMAX ATTENTION: COMBINING MRSHA WITH OTHER ATTENTIONS

The MrsHa is complementary to many other types of attentions. Therefore, a natural question arises is whether we can combine the MrsHa with other attentions besides the softmax attention? To answer this question, we combine the MrsHA with the MRA attention (Zeng et al., 2022) and the linear attention (Katharopoulos et al., 2020) and train these combined models for the LRA tasks (Tay et al.,

Table 5: Accuracy (%) of the models that combined MrsHa with the MRA and linear attentions vs. the original MRA and linear transformers on the LRA tasks. The combined models are indicated by the prefix "Mrs", results are averaged over 5 seeds (In this experiment, we use the set of scales $s = [1, 2]$).

| DATASET / MODEL | MRSMRA-2 (MRA-2) | MRSMRA-2-S (MRA-2-S) | MRSLINEAR (LINEAR) |
|---|---|---|---|
| LISTOPS | 37.05 (37.10) | 37.17 (37.05) | 36.97 (36.90) |
| RETRIEVAL | 79.24 (78.88) | 80.05 (79.76) | 81.36 (81.13) |
| TEXT | 64.92 (65.09) | 65.21 (64.43) | 66.57 (65.69) |
| AVERAGE ACCURACY | **60.40** (60.36) | **60.81** (60.42) | **61.63** (61.24) |

2021a) as in Section 3.2. The results are presented in Table 5. It is interesting to see from Table 5 that all combined models gain an improvement in average test accuracy over the original models despite being an approximation. This observation suggests that the MrsHa can be applied to other attention mechanisms besides softmax to reduce computation and memory while maintaining the accuracy of the original models.

## 5 RELATED WORK

**Efficient Transformers.** To reduce the quadratic computational cost and memory usage of transformers, many efficient transformer models have been developed (Roy et al., 2021). Sparse transformers are a line of works in this branch, which explore and design the sparsity structure of attention matrix, resulting in more efficient models (Parmar et al., 2018; Liu et al., 2018b; Qiu et al., 2019; Child et al., 2019; Beltagy et al., 2020). Another class of efficient transformers is patterns integration, combining different attention patterns to cover a diverse and wide range of dependencies (Child et al., 2019; Ho et al., 2019). These patterns can be set as pre-specified or learnable during training, along with model parameters (Kitaev et al., 2020; Roy et al., 2021; Tay et al., 2020). In another attempt, multiple tokens can be accessed simultaneously with a side memory module, saving the cost of computing and memory storage(Lee et al., 2019; Sukhbaatar et al., 2019; Asai & Choi, 2020; Beltagy et al., 2020). In a different approach, observing that the attention matrices are low-rank, kernelization and low-rank approximation methods have been proposed to replace the softmax attention with more efficient attentions (Tsai et al., 2019; Wang et al., 2020a; Katharopoulos et al., 2020; Choromanski et al., 2021; Shen et al., 2021; Nguyen et al., 2021; Peng et al., 2021; Jaegle et al., 2021). From a signal processing perspective, wavelet-based and multiscale methods has been used lately to learn a multiresolution approximation of self-attention (Zeng et al., 2022; Fan et al., 2021; Tao et al., 2020; Li et al., 2022), which flexibly discover the coarse and fine attention patterns. Our approach decomposes the attention heads into coarse- and fine-scale heads, diversely modeling the dependencies between tokens and between group of tokens to reduce the computational and memory costs of the model in both training and testing.

**Redundancy in Transformers.** Pre-trained transformers contain redundant neurons and heads which can be pruned away for downstream tasks (Dalvi et al., 2020; Michel et al., 2019; Durrani et al., 2020). Studying the contextualized embeddings in these pre-trained networks shows the anisotropicity of the learned representation from these models under this redundancy (Mu & Viswanath, 2018; Ethayarajh, 2019). Multiple approaches have been proposed to reduce this redundancy and improve the efficiency of transformers, such as the knowledge distillation and sparse approximation (Sanh et al., 2019; Sun et al., 2019; Voita et al., 2019b; Sajjad et al., 2020). Our MrsHA/MrsFormer represent the attention heads at different scales and are complementary to these methods.

## 6 CONCLUDING REMARKS

In this paper, we propose the MrsFormer, a class of efficient transformers that calculates the approximation of the attention heads at different scales using the Multiresolution-head Attention (MrsHA). The MrsFormer achieves better computational and memory cost than the corresponding softmax transformers baseline. Furthermore, the MrsFormer helps reduce the redundancy between attention heads and can be easily combined with other attention mechanisms. In the MrsFormer, we use the boxcar function to form a set of orthogonal expansion functions. It is natural to further develop the MrsFormer using other basis functions including the popular wavelets. Furthermore, in our derivation of the MrsHA and MrsFormer in Section 2.3, we employ the observation from (Nguyen et al., 2022) that the features $\mathbf{H}[:, d]$ in the output sequence $\mathbf{H}$ are independent. We leave the extenson of the MrsHA and MrsFormer to capture dependent output features as future work.

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

# Supplement to "MrsFormer: Transformer with Multiresolution-head Attention"

## A ADDITIONAL DETAILS ON THE EXPERIMENTS

### A.1 UEA TIME SERIES CLASSIFICATION

**Datasets and metrics** The benchmark (Bagnall et al., 2018) consists of 30 datasets. Following (Wu et al., 2022), we choose 10 datasets, which vary in input sequence lengths, the number of classes, and dimensionality, to evaluate our models on temporal sequences.

**Models and baselines** We adapt code from (Wu et al., 2022; Zerveas et al., 2021) for our experiments. Following the same setting from these papers, we set the number of heads and layers to 8 and 2, respectively. For the MrsFormers, we use the same set of scales at each layer, which is given by $s = [1, 1, 2, 2, 4, 4, 8, 8]$. For MRA-2 and MRA-2-s models (Zeng et al., 2022), each head is approximated by blocks of scales $[1, 32]$ as suggested in their paper. The percentage of blocks with scale 1 in these MRA-2 models is set to $25\%$ of the full attention matrix. Other hyperparameters have the same values as in (Wu et al., 2022) (for the PEMS-SF, SelfRegulationSCP2, and UWaveGestureLibrary tasks) and (Zerveas et al., 2021) (for other tasks). Hyperparameters for these tasks are presented in Table 6.

### A.2 LONG RANGE ARENA BENCHMARK

**Datasets and metrics** We adopt the tasks: Listops (Nangia & Bowman, 2018), byte-level IMDb reviews text classification (Maas et al., 2011), and byte-level document retrieval (Radev et al., 2013) in the LRA benchmark for our experiments. They consist of long sequences of length $2K$, $4K$, and $4K$, respectively. The evaluation protocol and metric are the same as in (Tay et al., 2021b).

**Models and baselines** We follow the same settings and adapt code for LRA task from (Zeng et al., 2022), which uses transformer with 2 heads and 2 layers. We choose the same set of scales $s = [1, 2]$ for all the layers in MsFormer. Hperparameters for these tasks are presented in Table 7.

### A.3 IMAGE CLASSIFICATION ON IMAGENET

**Dataset and metric:** We perform classification task on ILSVRC-2012 ImageNet dataset to validate the performance of our model on large dataset. This dataset has 1000 classes and about 1.28 million images.

**Models and baselines** In this section, we apply the MrsFormer to the Deit model (Touvron et al., 2020) with 4 heads. Since Deit uses special class token $[CLS]$ for the classification, we do not downsample this token along with other tokens in the sequence. For our MrsFormers, we use the set of scales $s = [1, 2, 2, 4]$ at each layer. We also study the MRA-2-s attention on this task. As reported in (Zeng et al., 2022), the MRA-2-s is a better model than the MRA-2 on the ImageNet image classification task since its sparse attention structure is more effective for modeling images.

## B PROOF OF THEOREM 1

Recall from Eqn. (14) that

$$\mathbf{H} \approx \mathbf{H}^{s,s'} = \uparrow_{s,1} ((\downarrow_{s,1}\downarrow_{s',2} \mathbf{A})(\downarrow_{s',1} \mathbf{V})).$$

Let $\mathbf{T}_s$ be the down-sampling operator (matrix multiplication) on the first dimension of a matrix corresponding to the scale $s$. $\mathbf{T}_s$ is the Kronecker product (or outer product) between an identity matrix $\mathbf{I}$ and the row vector $\frac{1}{s_i}\overrightarrow{\mathbf{1}}$ of size $1 \times s$, i.e. $\mathbf{T}_s = \mathbf{I} \otimes \frac{1}{s}\overrightarrow{\mathbf{1}}$. Under this notation, the up-sampling operator is the transpose of $\mathbf{T}_s$. In addition, the down-sampling operator on the second dimension of a matrix is also $\mathbf{T}_s^T$ but with the right multiplication instead. Then, we can rewrite the approximation $\mathbf{H}^{s,s'}$ as follows:

$$\mathbf{H}^{s,s'} = \mathbf{T}_s^T((\mathbf{T}_s\mathbf{A}\mathbf{T}_{s'}^T)(\mathbf{T}_{s'}\mathbf{V})) = (\mathbf{T}_s^T\mathbf{T}_s\mathbf{A}\mathbf{T}_{s'}^T\mathbf{T}_{s'})\mathbf{V}.$$

From the above equation, we have

$$\mathbf{H} - \mathbf{H}^{s,s'} = \left(\mathbf{A} - (\mathbf{T}_s^T\mathbf{T}_s\mathbf{A}\mathbf{T}_{s'}^T\mathbf{T}_{s'})\right)\mathbf{V}.$$

| Dataset | dim. model | dim. feedforward | learning rate | batchsize |
|---|---|---|---|---|
| SelfRegulationSCP2 | 512 | 2048 | 0.001 | 16 |
| PEMS-SF | 512 | 2048 | 0.001 | 16 |
| UWaveGestureLibrary | 512 | 2048 | 0.001 | 16 |
| EthanolConcentration | 64 | 256 | 0.001 | 128 |
| Handwriting | 128 | 256 | 0.001 | 128 |
| Heartbeat | 64 | 256 | 0.001 | 128 |
| JapaneseVowels size | 128 | 256 | 0.001 | 128 |
| SelfRegulationSCP1 size | 128 | 256 | 0.001 | 128 |
| SpokenArabicDigits size | 64 | 256 | 0.001 | 128 |
| FaceDetection size | 128 | 256 | 0.001 | 128 |

Table 6: Hyperparameter configuration for UEA time series classification task.

| Dataset | embedding dim | hidden dim | head dim | learning rate |
|---|---|---|---|---|
| listops | 64 | 128 | 32 | 0.0001 |
| retrieval | 64 | 128 | 32 | 0.0001 |
| text | 64 | 128 | 32 | 0.0001 |

Table 7: Hyperparameter configuration for LRA task.

From the inequality with the Frobenius norm, we have

$$\|\mathbf{H} - \mathbf{H}^{s,s'}\|_F \le \|\mathbf{A} - \mathbf{T}_s^T \mathbf{T}_s \mathbf{A} \mathbf{T}_{s'}^T \mathbf{T}_{s'}\|_F \|\mathbf{V}\|_2.$$

Therefore, it suffices to approximate the upper bound $\|\mathbf{A} - \mathbf{T}_s^T \mathbf{T}_s \mathbf{A} \mathbf{T}_{s'}^T \mathbf{T}_{s'}\|_F$. Let $\mathbf{A}^{s,s'} = \mathbf{T}_s^T \mathbf{T}_s \mathbf{A} \mathbf{T}_{s'}^T \mathbf{T}_{s'}$ and obviously $\mathbf{A}^{s,s'}$ contains blocks matrices of the same values. We can rewrite $\mathbf{A}$ and $\mathbf{A}^{s,s'}$ as block matrices of size $s \times s'$: $\mathbf{A} = [\mathbf{A}_{m,n}]_{m,n}$ and $\mathbf{A}^{s,s'} = [\mathbf{A}_{m,n}^{s,s'}]_{m,n}$ where $m = 0, 1, ..., \text{qlen}/s$, and $n = 0, 1, ..., \text{klen}/s'$. Note that all elements of $\mathbf{A}_{m,n}^{s,s'}$ have an identical value to the average of all elements of the sub-matrix $\mathbf{A}_{m,n}$.

Now we can decompose the above quantity into a sum of Frobenius norms:

$$\|\mathbf{A} - \mathbf{T}_s^T \mathbf{T}_s \mathbf{A} \mathbf{T}_{s'}^T \mathbf{T}_{s'}\|_F^2 = \sum_{m,n} \|\mathbf{A}_{m,n} - \mathbf{A}_{m,n}^{s,s'}\|_F^2.$$

Recall that from the hypothesis, we have

$$|\mathbf{A}_{i,j} - \mathbf{A}_{i\pm 1,j}| \le \delta, \ |\mathbf{A}_{i,j} - \mathbf{A}_{i,j\pm 1}| \le \delta. \tag{18}$$

Then, by applying Popoviciu's inequality, we have

$$\text{Var}[X] \le \frac{(M-m)^2}{4},$$

where $m = \inf X$ and $M = \sup X$. Since matrix is finite, the infimum and the maximum become the maximum and minimum respectively. By Assumption 18, we can approximate the upper bound of $M - m$ as follows:

$$(M-m)^2 \le (s + s' - 2)^2 \delta^2.$$

Integrate the sum, we find that

$$\|\mathbf{A} - \mathbf{A}^{s,s'}\|_F^2 \le \frac{\text{qlen}}{s} \frac{\text{klen}}{s'} (s + s' - 2)^2 \frac{\delta^2}{4}.$$

When we plug in $\text{klen} = \text{qlen} = N$, we obtain a simpler version:

$$\|\mathbf{A} - \mathbf{A}^{s,s'}\|_F \le \frac{s + s' - 2}{\sqrt{ss'}} \frac{N\delta}{2}.$$

As a consequence, we obtain the conclusion of the theorem.

## C  ADDITIONAL EXPERIMENTS

### C.1  COMBINING MRSHA WITH OTHER EFFICIENT ATTENTIONS

In this section, we combine the proposed MrsHA architecture with other efficient attention mechanisms to demonstrate MrsHA can be combined with other efficient transformer to reduce memory and computation requirements. We run our experiments on 5 efficient transformer including

Table 8: Accuracy (%) of the models that combined MrsHa with other efficient transformers versus the accuracy of the original efficient transformers on the UEA Time Series Classification task. The combined models are indicated by the prefix "Mrs", results are averaged over 5 seeds (In this experiment, we use the set of scales $s = [1, 1, 2, 2, 4, 4, 8, 8]$).

| DATASET / MODEL | MRSLINFORMER (LINFORMER) | MRSLINEAR (LINEAR) | MRSFMM (FMM) | MRSPERFORMER (PERFORMER) | MRSLUNA (LUNA) |
|---|---|---|---|---|---|
| ETHANOLCONCENTRATION | 32.70 (32.95) | 35.49 (34.35) | 34.47 (34.22) | 33.21 (33.59) | 33.71 (33.59) |
| FACEDETECTION | 68.83 (68.53) | 68.91 (68.46) | 69.53 (68.97) | 68.90 (68.96) | 68.64 (68.92) |
| HANDWRITING | 32.55 (32.47) | 32.98 (33.29) | 33.02 (31.57) | 30.51 (30.47) | 32.94 (32.32) |
| HEARTBEAT | 75.12 (75.12) | 75.45 (76.75) | 76.42 (75.77) | 75.61 (75.93) | 75.61 (75.77) |
| JAPANESEVOWELS | 98.56 (98.65) | 99.46 (99.28) | 99.64 (99.64) | 99.01 (99.19) | 99.46 (99.46) |
| PEMS-SF | 87.67 (86.51) | 83.43 (79.96) | 86.9 (82.47) | 84.59 (84.59) | 81.31 (81.12) |
| SELFREGULATIONSCP1 | 92.61 (91.47) | 91.13 (91.24) | 93.06 (92.26) | 91.13 (91.01) | 91.24 (90.78) |
| SELFREGULATIONSCP2 | 55.19 (57.41) | 54.26 (53.33) | 54.82 (54.44) | 54.44 (55.19) | 55.74 (55.37) |
| SPOKENARABICDIGITS | 98.91 (98.88) | 98.76 (98.86) | 99.48 (99.38) | 99.02 (98.84) | 99.03 (99.06) |
| UWAVEGESTURELIBRARY | 86.25 (85.62) | 82.19 (80.63) | 86.46 (85.73) | 85.10 (85.00) | 86.25 (87.08) |
| AVERAGE ACCURACY | **72.84** (72.76) | **72.21** (71.61) | **73.38** (72.45) | 72.15 (**72.28**) | **72.39** (72.35) |

Table 9: Accuracy (%) of models that combined MrsHa with other efficient transformers versus accuracy of the original efficient transformers (in the parentheses) in LRA task. The combined models are indicated by the prefix "Mrs", results are averaged over 5 seeds (In this experiment, we use the set of scales $s = [1, 2]$).

| DATASET / MODEL | MRSLINFORMER (LINFORMER) | MRSLINEAR (LINEAR) | MRSFMM (FMM) | MRSPERFORMER (PERFORMER) | MRSLUNA (LUNA) |
|---|---|---|---|---|---|
| LISTOPS | 36.93 (36.59) | 36.97 (36.90) | 37.77 (30.67) | 37.12 (36.41) | 37.03 (37.02) |
| RETRIEVAL | 78.38 (78.17) | 81.36 (81.13) | 81.65 (80.91) | 78.93 (78.67) | 74.54 (69.55) |
| TEXT | 57.39 (56.50) | 66.57 (65.69) | 68.39 (68.57) | 65.20 (65.17) | 64.51 (66.13) |
| AVERAGE ACCURACY | **57.57** (57.09) | **61.63** (61.24) | **62.60** (60.05) | **60.42** (60.08) | **58.69** (57.57) |

Table 10: The resutls of the comparison between MrsFT-Transformer and FT-Transformer. The ↑ symbol denotes that the metric being reported is accuracy (the higher the better), the ↓ symbol denotes that the metric being reported is root mean square error (the lower the better).

| DATASET / MODEL | FT-TRANSFORMER | MRSFT-TRANSFORMER |
|---|---|---|
| CALIFORNIA HOUSING ↓ | **0.4671** | 0.468 |
| ADULT INCOME ↑ | 85.76 | **85.87** |
| HELENA ↑ | 37.99 | **38.23** |
| JANNIS ↑ | 72.46 | **72.54** |
| HIGGS ↑ | **72.50** | 72.44 |
| ALOI ↑ | 95.48 | **95.52** |
| EPSILON ↑ | **89.65** | 89.58 |
| YEAR ↓ | 8.905 | **8.904** |
| COVERTYPE ↑ | 96.67 | **96.84** |
| YAHOO ↓ | **0.7567** | 0.7586 |
| MICROSOFT ↓ | 0.7474 | **0.7468** |

Linformer (Wang et al., 2020a), Linear transformer (Katharopoulos et al., 2020), FMM transformer (Nguyen et al., 2021), Performer (Choromanski et al., 2021) and Luna transformer (Ma et al., 2021). All experiments settings in this section follows directly from subsections 3.1 and 3.2 unless stated otherwise.

### C.1.1 UEA TIME SERIES CLASSIFICATION

Results in Table 8 presents the accuracy of the combined and original models on the UEA Time Series Classification task. All the efficient transformers in this experiment either maintain comparable performance or experience a boost in average accuracy when combined with MrsHA.

### C.1.2 LONG RANGE ARENA

In Listops experiments, we increase the number of training step from 5000 to 15000 to ensure convergence for all models. Table 9 further consolidates the advantage of the proposed MrsHA architecture. In fact, all the combined models obtain better average accuracy than the original models in the LRA task.

## C.2 Tabular data

We include a diverse set of 11 tabular dataset for our benchmarking: California Housing (Kelley Pace & Barry, 1997), Adult (Kohavi, 1996), Helena (Guyon et al., 2019), Jannis (Guyon et al., 2019), Higgs (Baldi et al., 2014), ALOI (Geusebroek et al., 2005), Epsilon (EP, simulated physics experiments), Year (Bertin-Mahieux et al., 2011), Covertype (Blackard & Dean, 1999), Yahoo (Chapelle & Chang, 2011), Microsoft (Qin & Liu, 2013). We follow all the train settings and use the default set of hyperparameters used in paper (Gorishniy et al., 2021) for all models. For simplicity, we omit the ensemble step from paper (Gorishniy et al., 2021). We report average accuracy over 5 random seed for both FT-Transformer (Gorishniy et al., 2021) and the combined model of MrsHA and FT-Transformer, which we denote MrsFT-Transformer.

Table 10 evidently shows that our combined model obtained better results in 7 over 11 tasks, while other tasks maintain comparable performance. This result consolidates the benefit of combining MrsHA with other transformer models in a diverse set of tasks.

## C.3 Efficiency when combining MrsHA with other efficient transformer

For illustration, we present FLOP and memory reduction ratios of train and test phases of our MrsFMM transformer comparing to the original FMM transformer for LRA retrieval task in Figure 5. Our model saves up to 35% of the original FLOP and has lower memory footprint, less than 65% and 85% of the original model for training and testing phases, respectively.

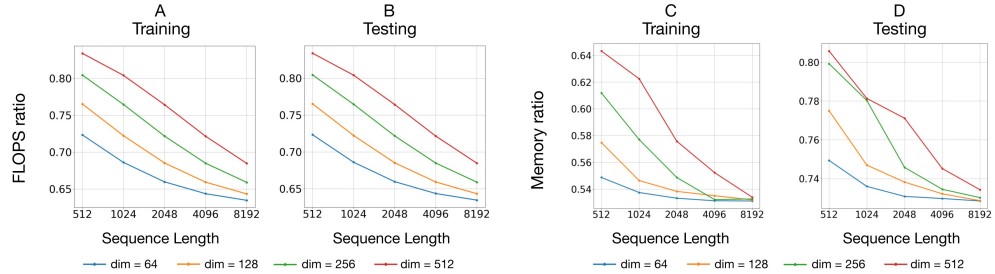

Figure 5: Training-inference FLOP ratios (A-B) and memory ratios (C-D) between the MrsFMM transformer and FMM transformer across different model dimensions and sequence lengths on the LRA retrieval task ($s = [1, 2]$).

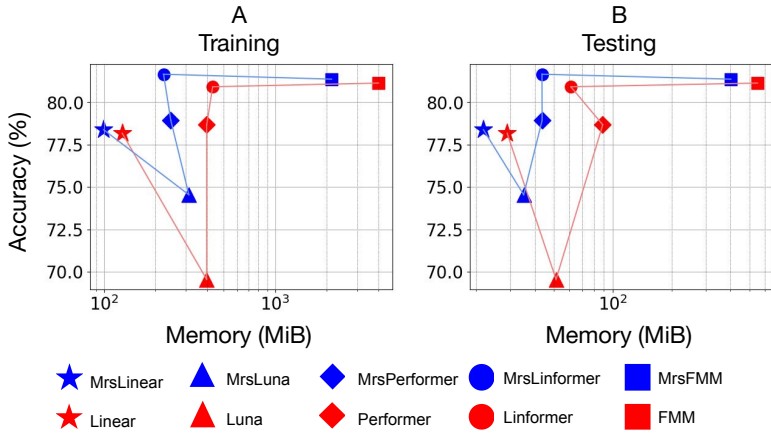

Figure 6: Scatter-plots for the relations between the memory usage and accuracy of the MrsHA-based efficient transformers vs. the baseline efficient transformers ($s = [1, 2]$) trained for the LRA retrieval task.

## C.4 Scatter-plots for the Relations between the Memory Usage and Accuracy of the MrsHA-based Efficient Transformers vs. the Baseline Efficient Transformers

We have included the scatter-plots for the relations between the memory usage and accuracy of the MrsLuna, MrsLinformer, MrsPerformer, MrsLinear, and MrsFMM vs. the Luna, Linformer,

Performer, Linear, and FMM baselines trained for the LRA retrieval task in Figure 6. We observe that in both train and test cases, the scatter-plots of our MrsHA-based models are above and on the left of the scatter-plots of the baselines, suggesting that our MrsHA-based models are more memory efficient while achieving comparable or better accuracies than the baseline models.

