# OpenReview forum: "Transformers with Multiresolution Attention Heads"
_ICLR.cc/2023/Conference — Submitted to ICLR 2023_

### Official Review · Reviewer_sjo4 · 2022-10-23

**Confidence:** 4
**Correctness:** 4
**Technical Novelty And Significance:** 2
**Empirical Novelty And Significance:** 2
**Recommendation:** 3

**Clarity, Quality, Novelty And Reproducibility:**

The technical discussions are clearly written. Many readers may find the paper easy-to-read
Thanks to the clear writings, reproducing may not be so hard.

I find two major concerns during the review.
(1) Novelty
Signal decomposition with orthogonal, multi-resolution bases passes the "test-of-time": no doubt in its efficacy.
If I correctly understand, the main idea is to simply plug-in the multi-reso decomposition with Haar-like Wavelet on the target matrices (Attention or V). I cannot find any other technical challenges or new ideas to solve the challenges.
Given these, it is difficult for me to evaluate the paper novel enough to accept at ICLR.

(2) impact of experiments
The experimental results are not strong.
The results show that the proposed method is marginally better than the existing Multi-Reso. Approximations. However, the main goal of the paper is to improve the efficiency of Transformer. In that sense, other approaches (not Multi-Reso Approx.) presented in the related work did better jobs.

For example, the FMMformer [Nguyen+, 2021] claims that the combination of low-rankness and sparsity yields linear-time complexity for self-attention.
NAFNet [Chen+, 2022] reduces the computational cost (in terms of Multiply-ACcumulation Operations MACs) up to 8.4% of the naive Transformer while achieving a new SotA in denoising.
Perceiver [Jaegle+, 2021] proposes to replace the pair-wise self-attention to the cross-attention between the low-dimensional Key latent vectors and the high-dimensional input vectors, yielding linear-time complexity with great recognition accuracy in several domains.

Compared to these results, the gains achieved in this paper are not significant. Then it is difficult for me to claim the necessity of the Multi-Res. Approx. approach to improve the efficiency of the Transformer at this moment.

I expect clearer advantages of the multi-res. approx. approch are provided through author-feedback time!


[Chen+, 2022] Chen, Chu, Zhang, and Sun, "Simple Baselines for Image Restoration", ECCV 2022.
[Jaegle+, 2021] Jaegle, Gimeno, Brock, Zisserman, Vinyals, and Carreira, "Perceiver: General Perception with Iterative Attention", ICML 2021.

Minor comments:
The discussion about the redundancy of the Multi-Reso Approximation in Sec 1.2: in general I agree to these claims, but it is better to have some citations to support the claims.

**Strength And Weaknesses:**

(+) Multi-resolution decomposition with orthogonal bases is a well-studied and proven-to-be-effective strategy for approximate signals. A reasonable choice.

(+) The paper reads easily, and straightforwardly. No logical jumps.

(--) Less technical novelty

(--) Less impact in experimental results


**Summary Of The Paper:**

This paper proposes a technique to improve the efficiency of multi-head attention.
The idea is to apply the multi-resolution decomposition with orthogonal bases inspired by wavelets to attention matrics or the V matrices.
The Haar Wavelet (box-car like) is considered in this paper, resulting in up/down sampling combined with multi-head attention.
Experimental results show that the proposed method improves FLOPS and memory uses compared to a naive softmax and the block-diagonal attention matrix approximation.

**Summary Of The Review:**

The technical idea is clear. But the novelty and the impact of experimental results are not enough for recommend to accept.

---

> ### Author Response · Authors · 2022-11-17
> **Response to Reviewer sjo4 (1)**
>
> Thank you for your thoughtful review and valuable feedback. Below we address your concerns.
>
> -----
> **Q1. Less technical novelty. Signal decomposition with orthogonal, multi-resolution bases passes the "test-of-time": no doubt in its efficacy. If I correctly understand, the main idea is to simply plug-in the multi-reso decomposition with Haar-like Wavelet on the target matrices (Attention or V). I cannot find any other technical challenges or new ideas to solve the challenges. Given these, it is difficult for me to evaluate the paper novel enough to accept at ICLR.**
>
> **Reply:**  We believe there is a misunderstanding of the novelty of our Transformer with Multiresolution-head Attention (MrsFormer). Please allow us to clear this misunderstanding by clarifying
> the main difference of our method compared to existing works and the key challenge of applying multiresolution approximation (MRA) to self-attention in our proposed method.
>
> First, multiresolution-head Attention (MrsHA) in the MrsFormer leverages the idea of the multiresolution approximation (MRA) to approximate the output of the multi-head attention (MHA). While existing works have been proposed to approximate the attention matrices or the value matrices V using the MRA [1,2,3,4], **our MrsHA is the first method that approximates the output $H$ of an attention head** given by Eqn. 1 in the paper, resulting in a better approximation scheme compared to other works that try to approximate the attention matrices. The approximation guarantee of the MrsHA is given in Theorem 1 in our paper.
>
> Second, applying the multi-resolution decomposition with Haar-like Wavelet on the output of an attention head is not simple. Wavelet decomposition was developed to approximate a given signal or matrix. The key challenge in approximating the output of an attention head is that this output is not given, but computed from the input to the attention layer via the formula given by Eqn. 1 in our paper. Section 2.3 in our revised manuscript (i.e. Section 2.2 in our original paper) derives an efficient way to approximate the attention output via downsampling the attention matrices and the value matrices followed by an upsampling step of the computed attention output (See Eqn. 13 in our revised manuscript). Even though the final formula of our MrsHA is simple, the derivation of this formula and the approximation guarantee of the MrsHA given in Theorem 1 are not trivial. This is the beauty of our work: We propose a principled method that works and is easy to implement but has deep theoretical justification and non-trivial derivation.
>
> **References**
>
> [1] Zeng, Zhanpeng, Sourav Pal, Jeffery Kline, Glenn M. Fung, and Vikas Singh. "Multi Resolution Analysis (MRA) for Approximate Self-Attention." In International Conference on Machine Learning, pp. 25955-25972. PMLR, 2022.
>
> [2] Fan, Haoqi, Bo Xiong, Karttikeya Mangalam, Yanghao Li, Zhicheng Yan, Jitendra Malik, and Christoph Feichtenhofer. "Multiscale vision transformers." In Proceedings of the IEEE/CVF International Conference on Computer Vision, pp. 6824-6835. 2021.
>
> [3] Tao, Andrew, Karan Sapra, and Bryan Catanzaro. "Hierarchical multi-scale attention for semantic segmentation." arXiv preprint arXiv:2005.10821 (2020).
>
> [4] Li, Yanghao, Chao-Yuan Wu, Haoqi Fan, Karttikeya Mangalam, Bo Xiong, Jitendra Malik, and Christoph Feichtenhofer. "MViTv2: Improved Multiscale Vision Transformers for Classification and Detection." In Proceedings of the IEEE/CVF Conference on Computer Vision and Pattern Recognition, pp. 4804-4814. 2022.
>
> -----

---

> > ### Author Response · Authors · 2022-11-17
> > **Response to Reviewer sjo4 (2)**
> >
> > **Q2. Impact of Experiments: The experimental results are not strong. The results show that the proposed method is marginally better than the existing Multi-Reso. Approximations. However, the main goal of the paper is to improve the efficiency of Transformer. In that sense, other approaches (not Multi-Reso Approx.) presented in the related work did better jobs. Compared to the results from these methods, the gains achieved in this paper are not significant. Then it is difficult for me to claim the necessity of the Multi-Res. Approx. approach to improve the efficiency of the Transformer at this moment.**
> >
> > **Reply:** Thanks for your comments. We would like to clarify that **our MrsHA is complementary to state-of-the-art efficient transformers** such as the FMMformer [1], NAFNet [2], Perceiver [3],  Luna [4], Linformer [5],  Performer [6], and Linear Transformer [7]. To further justify the advantage of MrsHA, we have combined MrsHA with Luna, Linformer, Performer, Linear Transformer, and FMMformer to obtain the MrsLuna, MrsLinformer, MrsPerformer, MrsLinear, and MrsFMM, respectively. Tables 8 and 9 in Appendix C of our revised manuscript show that MrsLuna, MrsLinformer, MrsPerformer, MrsLinear, and MrsFMM achieve better or at least comparable accuracies as the corresponding baselines on the UEA Time Series Classification task and the LRA tasks, respectively, while our methods are much more efficient (See Figure 5 in Appendix C.3 and Figures 3 and 4 in the main text of our revision).
> >
> > We have also combined our MrsHA with the FT-Transformer in [8]. Table 10 in our revision shows that our MrsFT-Transformer outperforms or is on par with the FT-Transformer baseline while being more efficient.
> >
> > Furthermore, we have conducted experiments on 2 additional tasks, ListOps and Text, in the LRA benchmarks to show that combining MrsHA with the MRA attention [9] helps improve this baseline MRA attention. We include the results in Table 5 in the main text of our revision.
> >
> > We have included NAFNet and Perceiver in the discussion in the Introduction and Related Work sections of our revision, respectively.
> >
> > **References**
> >
> > [1] Nguyen, Tan, Vai Suliafu, Stanley Osher, Long Chen, and Bao Wang. "Fmmformer: Efficient and flexible transformer via decomposed near-field and far-field attention." Advances in neural information processing systems 34 (2021): 29449-29463.
> >
> > [2] Chen, Liangyu, Xiaojie Chu, Xiangyu Zhang, and Jian Sun. "Simple baselines for image restoration." arXiv preprint arXiv:2204.04676 (2022).
> >
> > [3] Jaegle, Andrew, Felix Gimeno, Andy Brock, Oriol Vinyals, Andrew Zisserman, and Joao Carreira. "Perceiver: General perception with iterative attention." In International conference on machine learning, pp. 4651-4664. PMLR, 2021.
> >
> > [4] Ma, Xuezhe, Xiang Kong, Sinong Wang, Chunting Zhou, Jonathan May, Hao Ma, and Luke Zettlemoyer. "Luna: Linear unified nested attention." Advances in Neural Information Processing Systems 34 (2021): 2441-2453.
> >
> > [5] Wang, Sinong, Belinda Z. Li, Madian Khabsa, Han Fang, and Hao Ma. "Linformer: Self-attention with linear complexity." arXiv preprint arXiv:2006.04768 (2020).
> >
> > [6] Choromanski, Krzysztof, Valerii Likhosherstov, David Dohan, Xingyou Song, Andreea Gane, Tamas Sarlos, Peter Hawkins et al. "Rethinking attention with performers." arXiv preprint arXiv:2009.14794 (2020).
> >
> > [7] Katharopoulos, Angelos, Apoorv Vyas, Nikolaos Pappas, and François Fleuret. "Transformers are rnns: Fast autoregressive transformers with linear attention." In International Conference on Machine Learning, pp. 5156-5165. PMLR, 2020.
> >
> > [8] Gorishniy, Yury, Ivan Rubachev, Valentin Khrulkov, and Artem Babenko. "Revisiting deep learning models for tabular data." Advances in Neural Information Processing Systems 34 (2021): 18932-18943.
> >
> > [9] Zeng, Zhanpeng, Sourav Pal, Jeffery Kline, Glenn M. Fung, and Vikas Singh. "Multi Resolution Analysis (MRA) for Approximate Self-Attention." In International Conference on Machine Learning, pp. 25955-25972. PMLR, 2022.
> >
> > -----
> > **Q3. The discussion about the redundancy of the Multi-Reso Approximation in Sec 1.2: in general I agree to these claims, but it is better to have some citations to support the claims.**
> >
> > **Reply:** Thanks for your suggestion. We have added citations for the head redundancy issue in the multi-head attention (MHA) in the Introduction of our revised manuscript.
> >
> > -----
> >
> >
> > We hope we have cleared your concerns about our work. We have also revised our manuscript according to your comments, and we would appreciate it if we can get your further feedback at your earliest convenience.

---

> > > ### Comment · Reviewer_sjo4 · 2022-11-19
> > > **some comments**
> > >
> > > Authors,
> > >
> > > Thank you for the updates!
> > >
> > > * I think the motivation to approximate the H is now become clearer. Then let me ask another simple question. Reducing the number of heads does not solve the reduncancy problem?
> > >
> > > * I like the additional experiments on SotA Transformers in appendix. Personally, I personally feel this additional experiment results are more attractive than the results in the main mansucript.
> > >
> > > Currently, readers need to go back and forth between Tables 8.9 and Figure 5 to see the claim: "less computational budgets required while mainting the accuracy". This makes the experiment part less attractive.
> > >
> > > Just an idea: scatter-ploting the relations between memory and accuracy may ease this concern.
> > > Horizontal axis: memory usages, Vertial axis: accuracy, or whatever, performance scores. Plot the (mem, acc.) of Mrs-SotAs and SotAs. Mrs-SotAs should be plotted in left, while SotAs should be placed in right.
> > > Such a figure clearly presents the claim that the MrsTransformer cam improve the memory effciency without performance sacrifices.
> > >
> > > I'd like to determine my final score at the last moment, after exchanging with other reviewers given the updated manuscript. Thank you!

---

> > > > ### Author Response · Authors · 2022-11-19
> > > > **Response to Reviewer sjo4’s Comments**
> > > >
> > > > Thanks for your further feedback and valuable suggestions. We answer your questions below.
> > > >
> > > > -----
> > > > **Q1. Reducing the number of heads does not solve the redundancy problem?**
> > > >
> > > >
> > > > **Reply:** We believe there is a misunderstanding of our Multiresolution-head Attention (MrsHA). Please allow us to clear this misunderstanding by clarifying that our MrsHA does not reduce the number of heads. As shown Eqn. 16 in Definition 1 in our revised manuscript, our MrsHA reduces the computation and memory usage at each head by computing the approximation at scale $s^{(h)}$ of the output $\mathbf H^{(h)}$, $h=1,\dots,H$, of the $h^{th}$ head with an approximation of $\mathbf V^{(h)}$ at scale ${s^{'}}^{(h)}$. In other words, each attention head in MrsHA captures the attention output and thus the attention pattern at a different scale. In particular, the head at the finest scale in MrsHA captures the correlations between individual tokens in the sequence. The heads at the coarser scales in MrsHA capture the correlations between groups of tokens where the sizes of the groups depend on the approximation scales. As a result, attention heads in MrsHA capture different attention patterns, alleviating the head redundancy issue.
> > > >
> > > > **Q2. Readers need to go back and forth between Tables 8.9 and Figure 5 to see the claim: "less computational budgets required while maintaining the accuracy". This makes the experiment part less attractive. Scatter-ploting the relations between memory and accuracy may ease this concern**
> > > >
> > > > **Reply:** Following the reviewer’s suggestion, we have included the scatter-plots for the relations between the memory and accuracy of MrsLuna, MrsLinformer, MrsPerformer, MrsLinear, and MrsFMM vs. the Luna, Linformer, Performer, Linear, and FMM baselines trained for the LRA retrieval task in Figure 6 in Appendix C.4 of our revised manuscript. We are currently working on a similar figure for the UEA Time Series classification task and will include that figure in the discussion here.
> > > >
> > > > -----
> > > > We thank you again for your valuable feedback and encouragement.

---

> > > > > ### Author Response · Authors · 2022-11-25
> > > > > **Scatter-plots for the UEA Time Series Classification Task**
> > > > >
> > > > > As promised in our reply above, following the reviewer’s suggestion, we have included the scatter-plots for the relations between the memory and accuracy of MrsLuna, MrsLinformer, MrsPerformer, MrsLinear, and MrsFMM vs. the Luna, Linformer, Performer, Linear, and FMM baselines trained for the UEA Time Series classification task in Figure 2 in the following site:
> > > > >
> > > > > https://sites.google.com/view/transformer-multires-heads/home
> > > > >
> > > > > Figure 1 in the site above is a similar figure for the LRA retrieval task. This figure is the same as Figure 6 in Appendix C.4 of our revised manuscript.
> > > > >
> > > > > -----
> > > > > We thank you again for your valuable feedback and suggestions.

---

> > > > > > ### Author Response · Authors · 2022-11-27
> > > > > > **New Empirical Results on the ADE20K Semantic Segmentation Task.**
> > > > > >
> > > > > > Dear reviewer,
> > > > > >
> > > > > > We would like to thank the reviewer again for your thoughtful reviews and valuable feedback.
> > > > > >
> > > > > > We have conducted additional experiments on the ADE20K semantic segmentation task [1] to further confirm the advantage of our MrsHA in MrsFormer over baseline Softmax attention and the MRA-2-s attention [2]. The results in Table 1 below show that our MrsFormer improves over the baseline transformer with Softmax attention in both single-scale and multi-scale mean Intersection over Union (mIOU) while the MrsFormer is more efficient than the baseline. The MRA-2-s attention in [2] yields even worse results than the Softmax baseline. Here, all models consist of 14 transformer layers with 3 heads per layer, and the model dimension is 192. We follow the same model configurations and training settings as in [3].
> > > > > >
> > > > > > Table 1: The single-scale and multi-scale mean Intersection over Union (mIOU) of the MrsFormer vs. the transformer with the baseline Softmax attention and the MRA-2-s attention [2] on the ADE20K semantic segmentation task [1].
> > > > > >
> > > > > > | Model       | Single-scale mIOU        | Multi-scale mIOU   |
> > > > > > | :---        |    :----:   |    :----:   |
> > > > > > | Softmax     |   37.72 |    38.82   |
> > > > > > | MRA-2-s   |    36.24   |    37.37   |
> > > > > > | MrsFormer   |   **37.80**    |   **39.14**    |
> > > > > >
> > > > > > **References**
> > > > > >
> > > > > > [1] Bolei Zhou, Hang Zhao, Xavier Puig, Sanja Fidler, Adela Barriuso, Antonio Torralba. "Scene Parsing Through ADE20K Dataset." CVPR (2017).
> > > > > >
> > > > > > [2] Zeng, Zhanpeng, Sourav Pal, Jeffery Kline, Glenn M. Fung, and Vikas Singh. "Multi-Resolution Analysis (MRA) for Approximate Self-Attention." In International Conference on Machine Learning, pp. 25955-25972. PMLR, 2022.
> > > > > >
> > > > > > [3] Robin Strudel, Ricardo Garcia, Ivan Laptev, Cordelia Schmid. "Segmenter: Transformer for Semantic Segmentation". ICCV (2021).

---

### Official Review · Reviewer_VX4u · 2022-10-24

**Confidence:** 4
**Correctness:** 3
**Technical Novelty And Significance:** 2
**Empirical Novelty And Significance:** 2
**Recommendation:** 6

**Clarity, Quality, Novelty And Reproducibility:**

The topic is a very interesting and a challenging problem. And the proposed Multiresolution-head Attention is somehow novel. The performance shows the advantage of the proposed method. Overall, the paper is well-organized.

**Strength And Weaknesses:**

The proposed approach has higher computational efficiency and lower memory overhead than the original Transformer at the cost of a small amount of performance. Although the proposed method is more efficient than the current version of the vanilla Transformer, a comparison with the computational efficiency and memory cost of existing methods is missing.

**Summary Of The Paper:**

This paper explores the idea of using wavelet transform to compress the input data length to achieve the approximation of attention map and value matric in a Transformer. The experimental results show that it outperforms some state-of-the-art methods. And the proposed method performs better computational and memory cost than the vanilla Transformer.

**Summary Of The Review:**

The paper proposes an improved model of Transformer, Multiresolution-head Attention, to reduce computation and memory overhead. The proposed method is more efficient through additional experiments and complexity analysis than the original Transformer. However, the experimental part lacks a comparison with the algorithmic complexity of existing methods. The overall organization of the paper is clear.

---

> ### Author Response · Authors · 2022-11-17
> **Response to Reviewer VX4u (1)**
>
> Thank you for your thoughtful review and valuable feedback. Below we address your concerns.
>
> -----
> **Q1.  Although the proposed method is more efficient than the current version of the vanilla Transformer, a comparison with the computational efficiency and memory cost of existing methods is missing.**
>
> **Reply:** Thanks for your suggestion.  We would like to clarify that **our MrsHA is complementary to state-of-the-art efficient transformers** such as Luna [1], Linformer [2], and Performer [3]. To further justify the advantage of MrsHA, we have combined MrsHA with Luna, Linformer, Performer, Linear Transformer [4], and FMMformer [5] to obtain the MrsLuna, MrsLinformer, MrsPerformer, MrsLinear, and MrsFMM, respectively. Tables 8 and 9 in Appendix C of our revised manuscript show that MrsLuna, MrsLinformer, MrsPerformer, MrsLinear, and MrsFMM achieve better or at least comparable accuracies as the corresponding baselines on the UEA Time Series Classification task and the LRA tasks, respectively, while our methods are much more efficient (See Figure 5 in Appendix C.3 and Figures 3 and 4 in the main text of our revision).
>
> We have also combined our MrsHA with the FT-Transformer in [6]. Table 10 in our revision shows that our MrsFT-Transformer outperforms or is on par with the FT-Transformer baseline while being more efficient.
>
> Furthermore, we have conducted experiments on 2 additional tasks, ListOps and Text, in the LRA benchmarks to show that combining MrsHA with the MRA attention [7] helps improve this baseline MRA attention. We include the results in Table 5 in the main text of our revision.
>
>
> **References**
>
> [1] Ma, Xuezhe, Xiang Kong, Sinong Wang, Chunting Zhou, Jonathan May, Hao Ma, and Luke Zettlemoyer. "Luna: Linear unified nested attention." Advances in Neural Information Processing Systems 34 (2021): 2441-2453.
>
> [2] Wang, Sinong, Belinda Z. Li, Madian Khabsa, Han Fang, and Hao Ma. "Linformer: Self-attention with linear complexity." arXiv preprint arXiv:2006.04768 (2020).
>
> [3] Choromanski, Krzysztof, Valerii Likhosherstov, David Dohan, Xingyou Song, Andreea Gane, Tamas Sarlos, Peter Hawkins et al. "Rethinking attention with performers." arXiv preprint arXiv:2009.14794 (2020).
>
> [4] Katharopoulos, Angelos, Apoorv Vyas, Nikolaos Pappas, and François Fleuret. "Transformers are rnns: Fast autoregressive transformers with linear attention." In International Conference on Machine Learning, pp. 5156-5165. PMLR, 2020.
>
> [5] Nguyen, Tan, Vai Suliafu, Stanley Osher, Long Chen, and Bao Wang. "Fmmformer: Efficient and flexible transformer via decomposed near-field and far-field attention." Advances in neural information processing systems 34 (2021): 29449-29463.
>
> [6] Gorishniy, Yury, Ivan Rubachev, Valentin Khrulkov, and Artem Babenko. "Revisiting deep learning models for tabular data." Advances in Neural Information Processing Systems 34 (2021): 18932-18943.
>
> [7] Zeng, Zhanpeng, Sourav Pal, Jeffery Kline, Glenn M. Fung, and Vikas Singh. "Multi Resolution Analysis (MRA) for Approximate Self-Attention." In International Conference on Machine Learning, pp. 25955-25972. PMLR, 2022.
>
> -----
> We hope we have cleared your concerns about our work. We have also revised our manuscript according to your comments, and we would appreciate it if we can get your further feedback at your earliest convenience.

---

> > ### Author Response · Authors · 2022-11-27
> > **New Empirical Results on the ADE20K Semantic Segmentation Task. Any Further Questions from the Reviewer VX4u?**
> >
> > Dear reviewer,
> >
> > We would like to thank the reviewer again for your thoughtful reviews and valuable feedback.
> >
> > We have conducted additional experiments on the ADE20K semantic segmentation task [1] to further confirm the advantage of our MrsHA in MrsFormer over baseline Softmax attention and the MRA-2-s attention [2]. The results in Table 1 below show that our MrsFormer improves over the baseline transformer with Softmax attention in both single-scale and multi-scale mean Intersection over Union (mIOU) while the MrsFormer is more efficient than the baseline. The MRA-2-s attention in [2] yields even worse results than the Softmax baseline. Here, all models consist of 14 transformer layers with 3 heads per layer, and the model dimension is 192. We follow the same model configurations and training settings as in [3].
> >
> > Table 1: The single-scale and multi-scale mean Intersection over Union (mIOU) of the MrsFormer vs. the transformer with the baseline Softmax attention and the MRA-2-s attention [2] on the ADE20K semantic segmentation task [1].
> >
> > | Model       | Single-scale mIOU        | Multi-scale mIOU   |
> > | :---        |    :----:   |    :----:   |
> > | Softmax     |   37.72 |    38.82   |
> > | MRA-2-s   |    36.24   |    37.37   |
> > | MrsFormer   |   **37.80**    |   **39.14**    |
> >
> > **References**
> >
> > [1] Bolei Zhou, Hang Zhao, Xavier Puig, Sanja Fidler, Adela Barriuso, Antonio Torralba. "Scene Parsing Through ADE20K Dataset." CVPR (2017).
> >
> > [2] Zeng, Zhanpeng, Sourav Pal, Jeffery Kline, Glenn M. Fung, and Vikas Singh. "Multi-Resolution Analysis (MRA) for Approximate Self-Attention." In International Conference on Machine Learning, pp. 25955-25972. PMLR, 2022.
> >
> > [3] Robin Strudel, Ricardo Garcia, Ivan Laptev, Cordelia Schmid. "Segmenter: Transformer for Semantic Segmentation". ICCV (2021).
> >
> >
> > -----
> > We would appreciate it if you could let us know if there are additional questions or concerns about our revision and rebuttal. We would be happy to do any follow-up discussion or address any additional comments.

---

> > > ### Author Response · Authors · 2022-12-04
> > > **Any further questions on our current draft?**
> > >
> > > Dear reviewer,
> > >
> > > We would like to thank the reviewer again for your thoughtful reviews and valuable feedback. We have updated our manuscript and added new replies to your comments and questions with our latest experimental results.
> > >
> > > We would appreciate it if you could let us know if there are additional questions or concerns about our revision and rebuttal.
> > >
> > > Best regards,
> > >
> > > Authors

---

### Official Review · Reviewer_fFTA · 2022-10-27

**Confidence:** 3
**Correctness:** 3
**Technical Novelty And Significance:** 2
**Empirical Novelty And Significance:** 2
**Recommendation:** 3

**Clarity, Quality, Novelty And Reproducibility:**

The technical details are sufficient; however the experiments need include more state of the art methods as baselines to be more solid

**Strength And Weaknesses:**

Strengths:
* The authors focus on an important problem in sequence modeling
* The technical details are presented well

Weakness
* The novelty needs to be further justified
* The draft needs to be better organized
* The experiments need to be strengthen

**Summary Of The Paper:**

The authors propose a multi-resolution based attention mechanism named MrsFormer. Various experiments have been conducted to validate the effectiveness of the proposed method over several baselines

**Summary Of The Review:**

OnThe authors needs to better justify the novelty and the motivation of the proposed method. From the introduction, the authors only briefly introduce some technical description without any motivation or rationale of proposing MrsHA. I suggest the authors move the background knowledge parts (e.g., self-attention description) into Sec. 2 then highlight the motivation and justify the novelty.

The paper also misses many state-of-the-art efficient transformer variants such as Luna, Linformer, Performance, etc.. I suggest the authors include these relevant methods as baselines in the evaluation or better justify the criteria of why selecting the existing baseline methods.

---

> ### Author Response · Authors · 2022-11-17
> **Response to Reviewer fFTA (1)**
>
> Thank you for your thoughtful review and valuable feedback. Below we address your concerns.
>
> -----
> **Q1. The novelty needs to be further justified. The authors need to better justify the novelty and the motivation of the proposed method. From the introduction, the authors only briefly introduce some technical descriptions without any motivation or rationale of proposing MrsHA. I suggest the authors move the background knowledge parts (e.g., self-attention description) into Sec. 2 then highlight the motivation and justify the novelty.**
>
> **Reply:** Thanks for your comments. Following the reviewer’s suggestion, we have moved the background on self-attention to Section 2 and highlighted the motivation and novelty of our method in the Introduction. Please allow us to clarify the motivation and novelty of our Transformer with Multiresolution-head Attention (MrsFormer) here. As already explained in the introduction of our paper, attention heads in the multi-head attention (MHA) are redundant and tend to learn similar attention patterns, thus limiting the representation capacity of the model. Furthermore, additional heads increase the computational and memory costs, which becomes a bottleneck in scaling up transformers for very long sequences in large-scale practical tasks. These high computational and memory costs and head redundancy issues of the MHA motivates the need for a new efficient attention mechanism like the Multiresolution-head Attention (MrsHA) in the MrsFormer. MrsHA leverages the idea of the multiresolution approximation (MRA) to approximate the output of MHA. The MRA has been widely used to efficiently approximate complicated signals like video and images in signal and image processing [1,2,3], as well as to approximate solutions of partial differential equations [4,5]. While existing works have been proposed to approximate the attention matrices using the MRA [6,7,8,9], **our MrsHA is the first method that approximates the output $H$ of an attention head** given by Eqn. 1 in the paper, resulting in a better approximation scheme compared to other works that try to approximate the attention matrices. The approximation guarantee of the MrsHA is given in Theorem 1 in our paper.
>
>
> **References**
>
> [1] Mallat, Stéphane. A wavelet tour of signal processing. Elsevier, 1999.
>
> [2] JPEG2000 Image Compression: Fundamentals, Standards and Practice. Springer, 2002.
>
> [3] Bhaskaran, Vasudev, and Konstantinos Konstantinides. "Image and video compression standards: algorithms and architectures." (1997).
>
> [4] Dahmen, Wolfgang, Andrew Kurdila, and Peter Oswald. Multiscale wavelet methods for partial differential equations. Elsevier, 1997.
>
> [5] Qian, Sam, and John Weiss. "Wavelets and the numerical solution of partial differential equations." Journal of Computational Physics 106, no. 1 (1993): 155-175.
>
> [6] Zeng, Zhanpeng, Sourav Pal, Jeffery Kline, Glenn M. Fung, and Vikas Singh. "Multi Resolution Analysis (MRA) for Approximate Self-Attention." In International Conference on Machine Learning, pp. 25955-25972. PMLR, 2022.
>
> [7] Fan, Haoqi, Bo Xiong, Karttikeya Mangalam, Yanghao Li, Zhicheng Yan, Jitendra Malik, and Christoph Feichtenhofer. "Multiscale vision transformers." In Proceedings of the IEEE/CVF International Conference on Computer Vision, pp. 6824-6835. 2021.
>
> [8] Tao, Andrew, Karan Sapra, and Bryan Catanzaro. "Hierarchical multi-scale attention for semantic segmentation." arXiv preprint arXiv:2005.10821 (2020).
>
> [9] Li, Yanghao, Chao-Yuan Wu, Haoqi Fan, Karttikeya Mangalam, Bo Xiong, Jitendra Malik, and Christoph Feichtenhofer. "MViTv2: Improved Multiscale Vision Transformers for Classification and Detection." In Proceedings of the IEEE/CVF Conference on Computer Vision and Pattern Recognition, pp. 4804-4814. 2022.

---

> > ### Author Response · Authors · 2022-11-17
> > **Response to Reviewer fFTA (2)**
> >
> > **Q2. The experiments need to be strengthened. The paper also misses many state-of-the-art efficient transformer variants such as Luna, Linformer, Performance, etc. I suggest the authors include these relevant methods as baselines in the evaluation or better justify the criteria of why selecting the existing baseline methods.**
> >
> > **Reply:** Thanks for your suggestion.  We would like to clarify that **our MrsHA is complementary to state-of-the-art efficient transformers** such as Luna [1], Linformer [2], and Performer [3]. To further justify the advantage of MrsHA, we have combined MrsHA with Luna, Linformer, Performer, Linear Transformer [4], and FMMformer [5] to obtain the MrsLuna, MrsLinformer, MrsPerformer, MrsLinear, and MrsFMM, respectively. Tables 8 and 9 in Appendix C of our revised manuscript show that MrsLuna, MrsLinformer, MrsPerformer, MrsLinear, and MrsFMM achieve better or at least comparable accuracies as the corresponding baselines on the UEA Time Series Classification task and the LRA tasks, respectively, while our methods are much more efficient (See Figure 5 in Appendix C.3 and Figures 3 and 4 in the main text of our revision).
> >
> > We have also combined our MrsHA with the FT-Transformer in [6]. Table 10 in our revision shows that our MrsFT-Transformer outperforms or is on par with the FT-Transformer baseline while being more efficient.
> >
> > Furthermore, we have conducted experiments on 2 additional tasks, ListOps and Text, in the LRA benchmarks to show that combining MrsHA with the MRA attention [7] helps improve this baseline MRA attention. We include the results in Table 5 in the main text of our revision.
> >
> > **References**
> >
> > [1] Ma, Xuezhe, Xiang Kong, Sinong Wang, Chunting Zhou, Jonathan May, Hao Ma, and Luke Zettlemoyer. "Luna: Linear unified nested attention." Advances in Neural Information Processing Systems 34 (2021): 2441-2453.
> >
> > [2] Wang, Sinong, Belinda Z. Li, Madian Khabsa, Han Fang, and Hao Ma. "Linformer: Self-attention with linear complexity." arXiv preprint arXiv:2006.04768 (2020).
> >
> > [3] Choromanski, Krzysztof, Valerii Likhosherstov, David Dohan, Xingyou Song, Andreea Gane, Tamas Sarlos, Peter Hawkins et al. "Rethinking attention with performers." arXiv preprint arXiv:2009.14794 (2020).
> >
> > [4] Katharopoulos, Angelos, Apoorv Vyas, Nikolaos Pappas, and François Fleuret. "Transformers are rnns: Fast autoregressive transformers with linear attention." In International Conference on Machine Learning, pp. 5156-5165. PMLR, 2020.
> >
> > [5] Nguyen, Tan, Vai Suliafu, Stanley Osher, Long Chen, and Bao Wang. "Fmmformer: Efficient and flexible transformer via decomposed near-field and far-field attention." Advances in neural information processing systems 34 (2021): 29449-29463.
> >
> > [6] Gorishniy, Yury, Ivan Rubachev, Valentin Khrulkov, and Artem Babenko. "Revisiting deep learning models for tabular data." Advances in Neural Information Processing Systems 34 (2021): 18932-18943.
> >
> > [7] Zeng, Zhanpeng, Sourav Pal, Jeffery Kline, Glenn M. Fung, and Vikas Singh. "Multi Resolution Analysis (MRA) for Approximate Self-Attention." In International Conference on Machine Learning, pp. 25955-25972. PMLR, 2022.
> >
> > -----
> > **Q3. The draft needs to be better organized.**
> >
> > **Reply:** Thanks for your comments. We have re-organized our manuscript as the reviewer suggests.
> >
> > -----
> >
> > We hope we have cleared your concerns about our work. We have also revised our manuscript according to your comments, and we would appreciate it if we can get your further feedback at your earliest convenience.

---

> > > ### Author Response · Authors · 2022-11-27
> > > **New Empirical Results on the ADE20K Semantic Segmentation Task. Any Further Questions from Reviewer fFTA?**
> > >
> > > Dear reviewer,
> > >
> > > We would like to thank the reviewer again for your review and valuable feedback.
> > >
> > > We have conducted additional experiments on the ADE20K semantic segmentation task [1] to further confirm the advantage of our MrsHA in MrsFormer over baseline Softmax attention and the MRA-2-s attention [2]. The results in Table 1 below show that our MrsFormer improves over the baseline transformer with Softmax attention in both single-scale and multi-scale mean Intersection over Union (mIOU) while the MrsFormer is more efficient than the baseline. The MRA-2-s attention in [2] yields even worse results than the Softmax baseline. Here, all models consist of 14 transformer layers with 3 heads per layer, and the model dimension is 192. We follow the same model configurations and training settings as in [3].
> > >
> > > Table 1: The single-scale and multi-scale mean Intersection over Union (mIOU) of the MrsFormer vs. the transformer with the baseline Softmax attention and the MRA-2-s attention [2] on the ADE20K semantic segmentation task [1].
> > >
> > > | Model       | Single-scale mIOU        | Multi-scale mIOU   |
> > > | :---        |    :----:   |    :----:   |
> > > | Softmax     |   37.72 |    38.82   |
> > > | MRA-2-s   |    36.24   |    37.37   |
> > > | MrsFormer   |   **37.80**    |   **39.14**    |
> > >
> > > **References**
> > >
> > > [1] Bolei Zhou, Hang Zhao, Xavier Puig, Sanja Fidler, Adela Barriuso, Antonio Torralba. "Scene Parsing Through ADE20K Dataset." CVPR (2017).
> > >
> > > [2] Zeng, Zhanpeng, Sourav Pal, Jeffery Kline, Glenn M. Fung, and Vikas Singh. "Multi-Resolution Analysis (MRA) for Approximate Self-Attention." In International Conference on Machine Learning, pp. 25955-25972. PMLR, 2022.
> > >
> > > [3] Robin Strudel, Ricardo Garcia, Ivan Laptev, Cordelia Schmid. "Segmenter: Transformer for Semantic Segmentation". ICCV (2021).
> > >
> > >
> > > -----
> > > We would appreciate it if you could let us know if there are additional questions or concerns about our revision and rebuttal. We would be happy to do any follow-up discussion or address any additional comments.

---

> > > > ### Author Response · Authors · 2022-12-04
> > > > **Any further questions on our current draft?**
> > > >
> > > > Dear reviewer,
> > > >
> > > > We would like to thank the reviewer again for your reviews and valuable feedback. We have updated our manuscript and added new replies to your comments and questions with our latest experimental results.
> > > >
> > > > We would appreciate it if you could let us know if there are additional questions or concerns about our revision and rebuttal.
> > > >
> > > > Best regards,
> > > >
> > > > Authors

---

### Author Response · Authors · 2022-11-17
**General Response (1)**

Dear AC and reviewers,

Thanks for your thoughtful reviews and valuable comments, which have helped us improve the paper significantly.  We have updated our submission based on the reviewers' feedback, and we have highlighted our revision in blue.

Two main concerns from the reviewers are: (1) the novelty of our work needs to be justified and (2) more experimental results are needed . We address these concerns here.

(1) **Novelty:**  We believe there is a misunderstanding of the novelty of our Transformer with Multiresolution-head Attention (MrsFormer). Please allow us to clear this misunderstanding by clarifying
the main difference of our method compared to existing works and the key challenge of applying multiresolution approximation (MRA) to self-attention in our proposed method.

First, multiresolution-head Attention (MrsHA) in the MrsFormer leverages the idea of the multiresolution approximation (MRA) to approximate the output of the multi-head attention (MHA). While existing works have been proposed to approximate the attention matrices or the value matrices V using the MRA [1,2,3,4], **our MrsHA is the first method that approximates the output $H$ of an attention head** given by Eqn. 1 in the paper, resulting in a better approximation scheme compared to other works that try to approximate the attention matrices. The approximation guarantee of the MrsHA is given in Theorem 1 in our paper.

Second, applying the multi-resolution decomposition with Haar-like Wavelet on the output of an attention head is not simple. Wavelet decomposition was developed to approximate a given signal or matrix. The key challenge in approximating the output of an attention head is that this output is not given, but computed from the input to the attention layer via the formula given by Eqn. 1 in our paper. Section 2.3 in our revised manuscript (i.e. Section 2.2 in our original paper) derives an efficient way to approximate the attention output via downsampling the attention matrices and the value matrices followed by an upsampling step of the computed attention output (See Eqn. 13 in our revised manuscript). Even though the final formula of our MrsHA is simple, the derivation of this formula and the approximation guarantee of the MrsHA given in Theorem 1 are not trivial. This is the beauty of our work: We propose a principled method that works and is easy to implement but has deep theoretical justification and non-trivial derivation.

---

> ### Author Response · Authors · 2022-11-17
> **General Response (2)**
>
> (2) **Experimental Results:** We would like to clarify that **our MrsHA is complementary to state-of-the-art efficient transformers** such as Luna [5], Linformer [6], and Performer [7]. To further justify the advantage of MrsHA, we have combined MrsHA with Luna, Linformer, Performer, Linear Transformer [8], and FMMformer [9] to obtain the MrsLuna, MrsLinformer, MrsPerformer, MrsLinear, and MrsFMM, respectively. Tables 8 and 9 in Appendix C of our revised manuscript show that MrsLuna, MrsLinformer, MrsPerformer, MrsLinear, and MrsFMM achieve better or at least comparable accuracies as the corresponding baselines on the UEA Time Series Classification task and the LRA tasks, respectively, while our methods are much more efficient (See Figure 5 in Appendix C.3 and Figures 3 and 4 in the main text of our revision).
>
> We have also combined our MrsHA with the FT-Transformer in [10]. Table 10 in our revision shows that our MrsFT-Transformer outperforms or is on par with the FT-Transformer baseline while being more efficient.
>
> Furthermore, we have conducted experiments on 2 additional tasks, ListOps and Text, in the LRA benchmarks to show that combining MrsHA with the MRA attention [11] helps improve this baseline MRA attention. We include the results in Table 5 in the main text of our revision.
>
>
> **References**
>
> [1] Zeng, Zhanpeng, Sourav Pal, Jeffery Kline, Glenn M. Fung, and Vikas Singh. "Multi Resolution Analysis (MRA) for Approximate Self-Attention." In International Conference on Machine Learning, pp. 25955-25972. PMLR, 2022.
>
> [2] Fan, Haoqi, Bo Xiong, Karttikeya Mangalam, Yanghao Li, Zhicheng Yan, Jitendra Malik, and Christoph Feichtenhofer. "Multiscale vision transformers." In Proceedings of the IEEE/CVF International Conference on Computer Vision, pp. 6824-6835. 2021.
>
> [3] Tao, Andrew, Karan Sapra, and Bryan Catanzaro. "Hierarchical multi-scale attention for semantic segmentation." arXiv preprint arXiv:2005.10821 (2020).
>
> [4] Li, Yanghao, Chao-Yuan Wu, Haoqi Fan, Karttikeya Mangalam, Bo Xiong, Jitendra Malik, and Christoph Feichtenhofer. "MViTv2: Improved Multiscale Vision Transformers for Classification and Detection." In Proceedings of the IEEE/CVF Conference on Computer Vision and Pattern Recognition, pp. 4804-4814. 2022.
>
> [5] Ma, Xuezhe, Xiang Kong, Sinong Wang, Chunting Zhou, Jonathan May, Hao Ma, and Luke Zettlemoyer. "Luna: Linear unified nested attention." Advances in Neural Information Processing Systems 34 (2021): 2441-2453.
>
> [6] Wang, Sinong, Belinda Z. Li, Madian Khabsa, Han Fang, and Hao Ma. "Linformer: Self-attention with linear complexity." arXiv preprint arXiv:2006.04768 (2020).
>
> [7] Choromanski, Krzysztof, Valerii Likhosherstov, David Dohan, Xingyou Song, Andreea Gane, Tamas Sarlos, Peter Hawkins et al. "Rethinking attention with performers." arXiv preprint arXiv:2009.14794 (2020).
>
> [8] Katharopoulos, Angelos, Apoorv Vyas, Nikolaos Pappas, and François Fleuret. "Transformers are rnns: Fast autoregressive transformers with linear attention." In International Conference on Machine Learning, pp. 5156-5165. PMLR, 2020.
>
> [9] Nguyen, Tan, Vai Suliafu, Stanley Osher, Long Chen, and Bao Wang. "Fmmformer: Efficient and flexible transformer via decomposed near-field and far-field attention." Advances in neural information processing systems 34 (2021): 29449-29463.
>
> [10] Gorishniy, Yury, Ivan Rubachev, Valentin Khrulkov, and Artem Babenko. "Revisiting deep learning models for tabular data." Advances in Neural Information Processing Systems 34 (2021): 18932-18943.
>
> [11] Zeng, Zhanpeng, Sourav Pal, Jeffery Kline, Glenn M. Fung, and Vikas Singh. "Multi Resolution Analysis (MRA) for Approximate Self-Attention." In International Conference on Machine Learning, pp. 25955-25972. PMLR, 2022.
>
> -----
> We are glad to answer any further questions you have on our submission.

---

### Author Response · Authors · 2022-11-17
**Summary of Revision**

Incorporating the comments and suggestions from all reviewers, besides fixing typos and notations, we have made the following main changes in the revised paper.

1. We have combined MrsHA with Luna, Linformer, Performer, Linear Transformer, and FMMformer to obtain the MrsLuna, MrsLinformer, MrsPerformer, MrsLinear, and MrsFMM, respectively. We summarize our experimental results that show these combined models outperform the baseline models in Tables 8 and 9 in Appendix C of our revised manuscript.

2. We have added Figure 5 in Appendix C.3 to show the FLOP and memory reduction ratios of the train and test phases of our MrsFMM transformer compared to those of the original FMM transformer on the LRA retrieval task.

3. We have combined our MrsHA with the FT-Transformer. We summarize our experimental results that show this combined model outperforms the baseline FT-Transformer in Table 10 in Appendix C of our revised manuscript.

4. We have conducted experiments on 2 additional tasks, ListOps and Text, in the LRA benchmarks to show that combining MrsHA with the MRA attention helps improve this baseline MRA attention. We include the results in Table 5 in the main text of our revision.

5. We have moved the background on self-attention to Section 2 and highlighted the motivation and novelty of our method in the Introduction.

6. We have added citations for the head redundancy issue in the multi-head attention (MHA) in the Introduction of our revised manuscript.

7. We have added Tables 6 and 7 to Appendix A of our revised manuscript to provide hyperparameter configurations used in our UEA Time Series Classification and LRA experiments.

---

### Author Response · Authors · 2022-11-18
**Any Questions from the Reviewers before the Deadline to Update Our Draft?**

Dear reviewers,

We would like to thank all reviewers again for your thoughtful reviews and valuable feedback. We have updated our manuscript and added new replies to your comments and questions with our latest experimental results. We have summarized the changes we made in the manuscript in the Summary of Revision below.

We would appreciate it if you could let us know if there are additional questions or concerns about our revision and rebuttal.

Best regards,

Authors

---

> ### Comment · Reviewer_sjo4 · 2022-11-19
> **MHA?**
>
> Thank you for updated manuscript. Just a minor issue: "MHA" in the added paragraph in Sec. 1 is not defined until Sec. 2. As it looks similar to MRA, simply confusing..

---

> > ### Author Response · Authors · 2022-11-19
> > **MHA Stands for Multi-head Attention**
> >
> > Thanks for pointing this out. MHA in the added paragraph in Sec. 1 stands for Multi-head Attention. We have added an explanation for this in the revised manuscript that we have just re-submitted.
> >
> > We would be happy to discuss and address any additional comments from you and other reviewers.

---

### Author Response · Authors · 2022-11-27
**Any Questions from the Reviewers? New Empirical Results on the ADE20K Semantic Segmentation Task**

Dear reviewers,

We would like to thank all reviewers again for your thoughtful reviews and valuable feedback.

We have conducted additional experiments on the ADE20K semantic segmentation task [1] to further confirm the advantage of our MrsHA in MrsFormer over baseline Softmax attention and the MRA-2-s attention [2]. The results in Table 1 below show that our MrsFormer improves over the baseline transformer with Softmax attention in both single-scale and multi-scale mean Intersection over Union (mIOU) while the MrsFormer is more efficient than the baseline. The MRA-2-s attention in [2] yields even worse results than the Softmax baseline. Here, all models consist of 14 transformer layers with 3 heads per layer, and the model dimension is 192. We follow the same model configurations and training settings as in [3].

Table 1: The single-scale and multi-scale mean Intersection over Union (mIOU) of the MrsFormer vs. the transformer with the baseline Softmax attention and the MRA-2-s attention [2] on the ADE20K semantic segmentation task [1].

| Model       | Single-scale mIOU        | Multi-scale mIOU   |
| :---        |    :----:   |    :----:   |
| Softmax     |   37.72 |    38.82   |
| MRA-2-s   |    36.24   |    37.37   |
| MrsFormer   |   **37.80**    |   **39.14**    |

**References**

[1] Bolei Zhou, Hang Zhao, Xavier Puig, Sanja Fidler, Adela Barriuso, Antonio Torralba. "Scene Parsing Through ADE20K Dataset." CVPR (2017).

[2] Zeng, Zhanpeng, Sourav Pal, Jeffery Kline, Glenn M. Fung, and Vikas Singh. "Multi-Resolution Analysis (MRA) for Approximate Self-Attention." In International Conference on Machine Learning, pp. 25955-25972. PMLR, 2022.

[3] Robin Strudel, Ricardo Garcia, Ivan Laptev, Cordelia Schmid. "Segmenter: Transformer for Semantic Segmentation". ICCV (2021).


-----
We would appreciate it if you could let us know if there are additional questions or concerns about our revision and rebuttal. We would be happy to do any follow-up discussion or address any additional comments.

---

### Decision · Program_Chairs · 2023-01-20

**Decision:**

Reject

**Justification For Why Not Higher Score:**

- Limited novelty and experimental improvement compared to previous works in multi-resolution approximation of attention matrices
- Experiments needs consolidations and more analyses


**Justification For Why Not Lower Score:**

N/A

**Metareview: Summary, Strengths And Weaknesses:**

This paper introduces MrsFormer, a transformer network with multiresolution attention. The authors apply a multi-resolution decomposition to attention matrices and to the V matrices. Experiments are conducted for time series classification and for image classification on ImageNet. The authors show that MrsFormer can improve efficiency while limiting the drop in prediction performances.
The paper initially received two reject (3) recommendations and one borderline accept (6) recommendation. The main concerns pointed out by the reviewers related to the novelty of the approach, the significance in experiments and the lack of comparison to other efficient attention baselines. During the discussion period, the authors provided a combination of their approach with efficient attention methods, and new segmentation results on ADE-20K. Although these additional experimental results were appreciated by reviewers, they did not successfully answered to the concerns on novelty, and there remained a majority of reviewers voting for rejection after rebuttal.

The AC carefully reads the submission and discussions. Several works have recently explored the idea of a multi-resolution approximation of attention matrices (Zeng et al., 2022; Fan et al., 2021; Tao et al., 2020; Li et al., 2022), as discussed in the paper. The difference in the submission is to approximate the output of an attention head rather than the attention matrices, but the experimental results only show marginal improvements of the proposed approach. The AC also considers that the experiments can be consolidated. On ImageNet, the generalization performances should be assessed on ImageNet-v2 since the ImageNet dataset does not have a separate validation and test set. In addition, the proposed approach is inferior to the softmax baseline on ImageNet, but seems to be superior for time series classification or on the new segmentation results: the authors did not comment if their approach might have an expected regularization effect, or if the improvements in performances are due to other factors.
Therefore, the AC recommends rejection, but encourages the authors to re-submit their work based on the reviews' feedback.